# Labor Market Outcomes of People with HIV Pre- and Post-Diagnosis in the Netherlands

Andrei Tuiu [1] ✉, Esmée Zwiers[1], Wendy Janssens[2], Vita Jongen [3,4,5], Ard van Sighem [3,4], Ferdinand Wit [3,4], Menno Pradhan[1,2] & Marc van der Valk [3,4]

In the current era of effective antiretroviral therapy, HIV has become a manageable chronic condition. Little is known about the consequences of HIV on individuals' labor market outcomes. We study the impact of an HIV diagnosis using linked clinical data (the Dutch ATHENA cohort) and administrative data. A causal effect is estimated by comparing outcomes of people with HIV diagnosed between 2010 and 2022 ($n = 5960$) to a matched control group ($n = 59,600$) in a difference-in-difference design. We find that people with HIV are less likely to be employed, work fewer hours, earn less income, and are more likely to receive disability benefits up to 7 years after diagnosis. These effects are more pronounced for those diagnosed with late-stage HIV disease. Those with a non-late stage diagnosis experience a deterioration of socioeconomic outcomes, despite being less likely to experience clinically relevant symptoms at diagnosis. These findings highlight the need for continued efforts in prevention and early detection of HIV.

Human immunodeficiency virus (HIV) is transmitted primarily through sexual intercourse, blood contact, or from mother to child during pregnancy and/or breastfeeding. HIV targets the immune system, which if left untreated, makes a person vulnerable to opportunistic infections eventually leading to the development of acquired immunodeficiency syndrome (AIDS). The first person with HIV in the Netherlands was reported in 1982[1]. Due to medical innovations, HIV has transformed from a deadly to a chronic condition over the past decades. The development and roll-out of effective combination antiretroviral therapy (ART) has significantly improved both the clinical outcomes and life expectancy for people with HIV. The life expectancy of individuals diagnosed and treated relatively early approaches that of the general population[2–4].

However, little is known about the impact of receiving an HIV diagnosis on quality of life in a general sense, including an individual's ability to participate in the labor market. An HIV diagnosis may impact individual labor market outcomes through possible health complications resulting from the HIV infection[4], a potential impact on mental health[5–7], and stigmatization and discrimination in the labor market[8,9], but individuals may also adjust their labor-leisure trade-off in response to a HIV-related health shock[10,11].

Previous studies show that HIV is associated with poorer labor market outcomes, as people with HIV are less likely to be employed compared to the general population[12–15]. Importantly, these studies are correlational in nature, meaning that it is unclear if observed differences in socioeconomic outcomes that occur after HIV acquisition or diagnosis are caused by HIV itself or by confounders. In addition, some of these studies rely on small survey samples with limited follow-up, attrition, or selective non-response[12,13].

We estimate the causal effect of receiving an HIV diagnosis on individual-level labor market outcomes. We combine clinical data on a census of people linked to HIV care in the Netherlands with administrative data on demographics and labor market outcomes of all residents. The data include information on employment, hours worked, income, and the receipt of disability benefits. We match people with HIV diagnosed between 2010 and 2022 to a control group of people

[1]Amsterdam School of Economics, University of Amsterdam, Amsterdam, the Netherlands. [2]School of Business and Economics, Vrije Universiteit Amsterdam, Amsterdam, the Netherlands. [3]Stichting hiv monitoring, Amsterdam, the Netherlands. [4]Amsterdam University Medical Center, University of Amsterdam, Department of Infectious Diseases, Amsterdam Infection & Immunity Institute, Amsterdam, the Netherlands. [5]Department of Infectious Diseases, Public Health Service Amsterdam, Amsterdam, the Netherlands. ✉e-mail: a.tuiu@uva.nl

without HIV from the Dutch population on registered sex, birth year, education level and migration background. We estimate a staggered difference-in-differences model that uses a 14-year bandwidth around the date of diagnosis. This method estimates a causal effect by comparing labor market outcomes of people with HIV before and after diagnosis to the development of labor market outcomes of matched controls. The causal effect is identified under two main assumptions, namely the parallel trends assumption and the no anticipation assumption. The former states that the outcomes of individuals with HIV and those without would have evolved in parallel absent an HIV diagnosis. The latter states that individuals do not anticipate receiving an HIV diagnosis.

By identifying the causal effects of receiving an HIV diagnosis on labor market outcomes, we highlight the need for continued efforts of prevention and early detection of HIV, as well as for targeted interventions aimed at helping people with HIV reintegrate in the labor market. The Netherlands is a high-income country with universal healthcare access including ART[16], a strong social support system and relatively low health inequalities. Therefore, the Netherlands likely represents a "best-case" scenario in which changes in labor market outcomes after an HIV diagnosis cannot be explained by insufficient access to relevant healthcare and social services. The impact of receiving an HIV diagnosis on labor market outcomes may be larger in settings with less social support and poorer access to care.

## Results

### Baseline characteristics

As described in the Methods section, we select a sample of 5960 people diagnosed with HIV and match it to a control group of 59,600 members of the general Dutch population. Of the people with HIV, 3246 had a non-late (i.e., timely) diagnosis, 2399 a late diagnosis and 315 could not be assigned to either group. Characteristics of the people diagnosed with HIV, subdivided by disease stage at the time of diagnosis (late and non-late diagnosis), the general Dutch population, and controls not diagnosed with HIV are shown in Table 1. Panel A shows individuals' demographic characteristics. Panel B shows labor market outcomes in the year prior to diagnosis for people with HIV, and at the same age for matched controls. Panel C reports the clinical characteristics of people with HIV by disease stage at the time of diagnosis. By definition, individuals with a late-stage diagnosis have a lower median CD4 count – 130 (IQR: 40–240) cells/mm³ compared to 530 (IQR: 413–690) cells/mm³ - and a higher probability of presenting with AIDS-defining conditions around the time of diagnosis (18.4 percent versus 0.3 percent) compared to people with a non-late diagnosis. The median time between diagnosis and ART initiation was 23 (IQR: 14– 40) days for individuals with a late-stage diagnosis, versus 52 (IQR: 24–299) days for individuals with a non-late diagnosis. Treatment is generally effective, as 97.2 percent of individuals in either group are virally suppressed (i.e. having plasma HIV-1 RNA levels below 200 copies/mL) within one year of initiating ART.

### Employment

Figure 1a shows the proportion of people with HIV and controls who are employed for at least one month in a given year. The outcomes of matched controls are captured at the same age as their matched person with HIV. Both groups experience a downward trend in employment, likely due to individuals reaching (early) retirement age, experiencing work-limiting disabilities over their careers, or otherwise exiting the labor market. These trends in employment run in parallel between the two groups in the pre-diagnosis period, but after diagnosis, a large decline in employment is observed for people with HIV relative to matched controls. In the year before diagnosis, people with HIV are 1.4 percentage points less likely to be employed than matched controls. This difference increases to 4.3 percentage points in the year after diagnosis.

Figure 1b distinguishes employment by HIV disease stage at diagnosis. Pre-diagnosis, the share of individuals in employment evolves in parallel between people with HIV and matched controls regardless of the disease stage at diagnosis. The reduction in employment compared to matched controls after diagnosis is more pronounced for the late-diagnosed group. One year after diagnosis, 62.8 percent of late-diagnosed people with HIV are employed versus 68.7 percent of the matched controls. This is 70 percent versus 73.1 percent in the non-late group. The prediagnosis differences in employment between the late and non-late diagnosed people with HIV, as well as their associated controls, might be explained by demographic differences between the groups, as shown in Table 1, such as the fact that the median age at diagnosis for individuals diagnosed late is higher than for individuals not diagnosed late (45 years compared to 41 years). Additionally, the difference could be caused by unobserved characteristics, such as occupational opportunities and choices. It is likely that pre-diagnosis differences between people with HIV and controls predate the infection with HIV, as a majority of people with HIV are estimated to be diagnosed within seven years of an infection[17].

Figure 1c formalizes these differences in an event study, showing point estimates and 95% confidence intervals over time relative to diagnosis. Point estimates are not significantly different from zero before diagnosis, which implies that employment develops similarly for people with HIV and matched controls. Point estimates become significantly negative after diagnosis, which shows that people with HIV experience a decline in employment after diagnosis. These results are also summarized in Table 2 where the pre- and post-diagnosis point estimates are aggregated into one estimate (the average treatment effect on the treated, ATT). An HIV diagnosis leads to an employment decline of 2.8 percentage points, which is a 3.9 percent reduction compared to mean employment prior to diagnosis.

Figure 1d shows the event study estimates separately for the late- and non-late diagnosed groups. For both groups, point estimates are not significantly different from zero before diagnosis. Point estimates become significantly negative after diagnosis. People with HIV experience a decline in employment starting within one year after diagnosis regardless of the stage of the disease at diagnosis. The effects are more pronounced for people with HIV who are diagnosed late, who experience a persistent and larger decline in employment starting in the year of diagnosis. However, late diagnosed individuals also experience a narrowing of the gap over time, which might be the result of selective attrition (unhealthier individuals leaving the sample due to death) and/or of an improvement in health following the initiation of antiretroviral therapy. Following diagnosis, employment declines by 1.9 percentage points, or 2.6 percent of the pre-diagnosis mean in the non-late diagnosed group, and 4.4 percentage points, or 6.3 percent of the pre-diagnosis mean in the late-diagnosed group, as shown in Table 2.

### Hours worked

Figure 2a shows work hours for people with HIV and matched controls relative to diagnosis. Data on hours worked are only available starting in 2006, which restricts the number of years before diagnosis that we can observe to four. Hours worked develop in parallel for people with HIV and matched controls before diagnosis and diverge after diagnosis. In the year before diagnosis, people with HIV work on average 0.65 FTE compared to 0.69 FTE for matched controls. Work hours decline to 0.61 FTE for people with HIV in the year after diagnosis. Figure 2b distinguishes by disease stage at diagnosis. Trends in hours worked are similar before diagnosis for the late and non-late diagnosed groups. Post-diagnosis, people with HIV in the late group experience a larger decline in work hours compared to the non-late group.

Figure 2c reports the event study estimates. Point estimates are not statistically different from zero before diagnosis. After diagnosis we observe a decline in FTE for people with HIV compared to matched

## Table 1 | Summary Statistics Table

| | People with HIV overall | Matched controls overall | People with HIV non-late diagnosis | Matched controls non-late diagnosis | People with HIV late diagnosis | Matched controls late diagnosis | General population unmatched |
|---|---|---|---|---|---|---|---|
| | (1) | (2) | (3) | (4) | (5) | (6) | (7) |
| Observations | 5960 | 59,600 | 3246 | 32,460 | 2399 | 23,990 | 176,287 |
| *Panel A: Demographics* | | | | | | | |
| Age [years] | 42 (35–50) | 42 (35–50) | 41 (33–48) | 41 (33–48) | 45 (37–52) | 45 (37–52) | 39 (18–57) |
| Male [%] | 87.6 | 87.6 | 89.8 | 89.8 | 85.5 | 85.5 | 49.4 |
| Education | | | | | | | |
| Low education [%] | 36.3 | 36.3 | 35.7 | 35.7 | 36.1 | 36.1 | 46.9 |
| Medium education [%] | 29.0 | 29.0 | 30.4 | 30.4 | 27.7 | 27.7 | 13.4 |
| High education [%] | 13.5 | 13.5 | 14.5 | 14.5 | 12.6 | 12.6 | 6.7 |
| Unknown education [%] | 21.2 | 21.2 | 19.4 | 19.4 | 23.6 | 23.6 | 33.0 |
| Origin | | | | | | | |
| Native Dutch [%] | 66.4 | 66.4 | 69.9 | 69.9 | 62.0 | 62.0 | 78.8 |
| 1st generation migrant [%] | 23.9 | 23.9 | 20.1 | 20.1 | 28.9 | 28.9 | 10.5 |
| 2nd generation migrant [%] | 9.7 | 9.7 | 10.0 | 10.0 | 9.1 | 9.1 | 10.7 |
| *Panel B: Mean outcomes* | | | | | | | |
| Employment [%] | 71.9 | 73.3 | 73.6 | 75.0 | 69.9 | 70.9 | |
| Work hours relative to FTE [% of FTE] | 65.0 | 69.5 | 67.3 | 71.5 | 62.2 | 66.8 | |
| Primary income [euros] | 35,696.7 | 44,237.7 | 38,602.6 | 45,496.0 | 32,212.7 | 42,845.8 | |
| Disability insurance take-up [%] | 7.8 | 7.1 | 7.6 | 6.3 | 8.1 | 8.3 | |
| *Panel C: Clinical data* CD4 at diagnosis [cells/mm$^3$] | 367.5 (160–570) | | 530 (413–690) | | 130 (40–240) | | |
| Time to ART initiation [days] | 34 (17–105) | | 52 (24–299) | | 23 (14–40) | | |
| Virally suppressed within 1 year of ART [%] | 97.2 | | 97.2 | | 97.3 | | |
| AIDS event [%] | 7.6 | | 0.03 | | 18.4 | | |

Notes: This table shows summary statistics for people with HIV and controls, overall and by stage at diagnosis. Column (1) refers to the total sample of people with HIV. Columns (2) and (3) stratify by the stage of the infection at diagnosis, non-late and late. This is based on the CD4 cell count at diagnosis, on previous negative tests and on the presence of AIDS-defining conditions. Column (4) refers to the entire matched control group. Columns (5) and (6) stratify individuals in the control group by the stage at diagnosis of their respective matched person with HIV. Column (7) refers to a random sample of unmatched individuals from the general Dutch population. Data in column (7) is measured in the year 2010. Panel A shows demographic characteristics in the year before diagnosis. Age is shown as median (IQR). Low education is defined as having completed pre-vocational secondary education (VMBO) and/or first three years of senior general secondary education (HAVO) or pre-university level (VWO). Medium education is defined as having completed secondary vocational education (MBO), senior general secondary education ('HAVO') or pre-university level (VWO). High education is defined has having completed higher vocational education (HBO) or university. Being native Dutch indicates a person born in the Netherlands to parents born in the Netherlands. 1st generation migrant indicates a person born abroad, with at least one parent born abroad. 2nd generation migrant indicates a person born in the Netherlands, with at least one parent born abroad. Panel B shows mean outcomes for each group, measured in the year before diagnosis. Employment is defined as having income from employment for at least one month in a given year. Hours worked is defined in full-time equivalents (FTE), which implies that a value of one represents a full-time job, and zero that the individual does not work. Income is defined as gross annual income from work and self-employment and the median is reported. Disability insurance (DI) take-up is defined as receiving disability benefits for at least one month in a given year. Panel C shows key clinical variables of people with HIV. CD4 at diagnosis measures the CD4 cell count of individuals at the time of diagnosis in cells/mm$^3$ of blood, and it is reported as median (IQR). Time to ART initiation is the time in days between diagnosis and when treatment is initiated, reported as median (IQR). Virally suppressed within 1 year of ART is an indicator for the individual achieving viral suppression within one year of initiating ART. It is defined as having fewer than 200 copies of HIV per milliliter of blood. AIDS event denotes whether the individual has had an AIDS-related event before and up to the time of diagnosis.

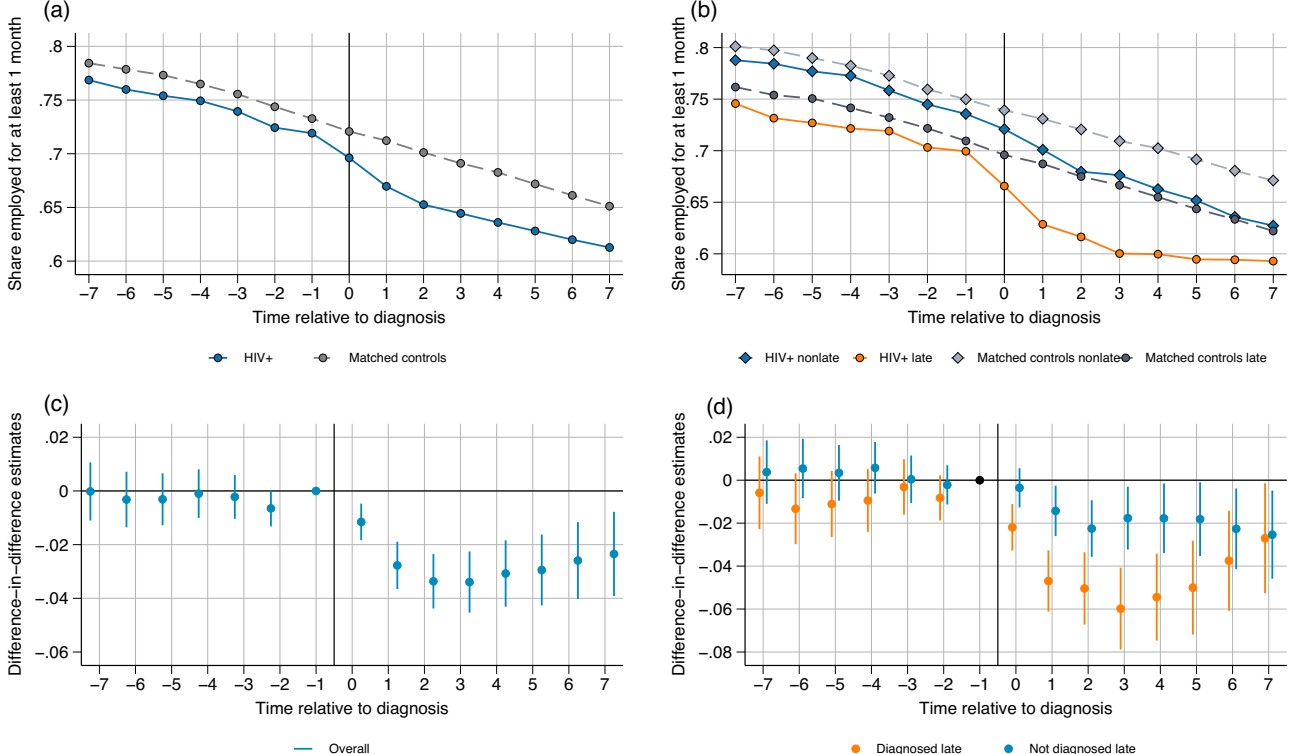

**Fig. 1 | Employment.** Employment is defined as having income from employment for at least one month in a given year. Panel (**a**) People with HIV and matched controls shows the proportion of employed individuals over time relative to diagnosis (at t = 0) for people with HIV (solid blue line) and matched controls (dashed gray line). Panel (**b**) People with HIV and controls by disease stage makes an additional distinction between people with HIV and matched controls depending on stage at diagnosis: non-late (solid blue line and dashed light gray line, respectively) and late (solid orange line and dark gray dashed line, respectively). Panels (**c**) Difference-in-differences estimates and (**d**) Difference-in-differences estimates by stage report event studies estimated using the Callaway and Sant'Anna estimator[50]. Each dot represents a point estimate for a specific period relative to diagnosis, and the bars represent the 95% confidence intervals, where t = 0 represents the year of diagnosis. In panel (**c**), N = 65,560 observations at period t-1. In panel (**d**), N = 35,706 observations at period t-1 for individuals not diagnosed late and N = 26,389 observations at period t-1 for individuals diagnosed late. The period t = −1 is the reference period and is standardized to zero. Source data are provided as a Source Data file.

controls that persists up to seven years. Table 2 shows that people with HIV experience a reduction in work hours of 3.5 percent of FTE after diagnosis, or 5.4 percent compared to their pre-diagnosis mean. For both the late and non-late diagnosis group, point estimates are not statistically different from zero prior to diagnosis, and show a decline starting in the year of diagnosis (Fig. 2d). The reduction is more pronounced for the late-diagnosed group, although this difference is not statistically significant. Table 2 shows that hours worked decline by 3.9 percent of FTE for individuals diagnosed late, a 6.3 percent reduction from the pre-diagnosis mean, and 3.4 percent of FTE for individuals not diagnosed late, a 5.1 percent reduction from the pre-diagnosis mean.

**Income**

Figure 3a shows yearly income from work or self-employment in euros over time relative to diagnosis for people with HIV and matched controls. The income of individuals who are not working is set to zero. The pre-diagnosis trend for people with HIV is somewhat flatter than for the matched controls. However, as shown in Fig. 3c, the pre-diagnosis estimates are not statistically different from zero. The mean yearly income of people with HIV in the year before diagnosis is 40,135 euros, compared to 48,096 euros for the matched controls. This large difference, which exists from at least seven years before diagnosis, might be explained by unobserved characteristics that we do not control for, such as different occupational choices and opportunities between the two groups. In the year of diagnosis, mean income of people with HIV declines by 1492 euros, and does not recover to the pre-diagnosis level

in subsequent years. Figure 3b shows mean income separately for the late- and non-late diagnosed groups. Both groups experience large reductions in yearly income at the time of diagnosis. The annual income of the non-late diagnosed group declines from 42,699 euros in the year before diagnosis to 41,084 in the year of diagnosis, a decline of 1615 euros. The income of the late diagnosed group declines from 37,451 to 35,857, a reduction of 1594 euros a year.

Point estimates and 95% confidence intervals resulting from comparing people with HIV and matched controls show no difference in the pre-diagnosis period (Fig. 3c). Post-diagnosis, we observe a large and persistent decline in yearly income. Table 2 shows that people with HIV experience a reduction in yearly income of 3583.8 euros after diagnosis, which is equivalent to 8.9 percent of the pre-diagnosis mean. Both people with late and non-late diagnosis experience persistent reductions in yearly income (Fig. 3d). Late-diagnosed individuals experience a decline of 3220.3 euros, or 8.6 percent of the pre-diagnosis mean, while non-late diagnosed individuals experience a loss of 3936.5 euros, or 9.2 percent of the pre-diagnosis mean (Table 2). These estimates are not statistically significantly different from one another given that the 95% confidence intervals for late- and non-late diagnosed individuals overlap in Fig. 3 and Table 2. Hence, people with HIV experience similar reductions in absolute income regardless of the HIV disease stage at diagnosis, which might be explained by the fact that individuals with a timely diagnosis are more likely to be employed at baseline, implying larger potential losses: 18.2% of individuals diagnosed late have no income one year before diagnosis, compared to 14.4% of individuals not diagnosed late.

**Table 2 | Main results: Average Treatment Effects on the Treated**

| | All people with HIV (1) | People with HIV not diagnosed late (2) | People with HIV diagnosed late (3) |
|---|---|---|---|
| *Panel A: Employment* ATT | −0.028*** | −0.019** | −0.044*** |
| | (0.005) | (0.006) | (0.008) |
| Pre-diagnosis mean | 0.719 | 0.736 | 0.699 |
| Effect size | −3.89% | −2.58% | −6.29% |
| Observations | 65,560 | 35,706 | 26,389 |
| *Panel B: Work hours* ATT | −0.035*** | −0.034*** | −0.039*** |
| | (0.005) | (0.006) | (0.007) |
| Pre-diagnosis mean | 0.650 | 0.673 | 0.622 |
| Effect size | −5.38% | −5.05% | −6.27% |
| Observations | 61,974 | 33,792 | 24,937 |
| *Panel C: Income* ATT | −3583.8*** | −3936.5*** | −3220.3*** |
| | (552.7) | (875.9) | (601.7) |
| Pre-diagnosis mean | 40,135.2 | 42,699.7 | 37,451.9 |
| Effect size | -8.93% | -9.22% | -8.59% |
| Observations | 64,416 | 35,101 | 25,916 |
| *Panel D: Disability Insurance* ATT | 0.036*** | 0.028*** | 0.049*** |
| | (0.004) | (0.004) | (0.006) |
| Pre-diagnosis mean | 0.078 | 0.076 | 0.081 |
| Effect size | 46.15% | 36.84% | 60.49% |
| Observations | 65,560 | 35,706 | 26,389 |

Notes: This table summarizes the event-study estimates in Figs. 1–4 by aggregating them into a before-after estimate, i.e., the effect of an HIV diagnosis. Average treatment effects on the treated (ATT) are estimated using the Callaway-Sant'Anna estimator and aggregated according to the procedure for staggered difference-in-differences models[50]. Column (1) shows the ATT for people with HIV, column (2) shows the ATT for people with HIV not diagnosed late and column (3) shows the ATT for people with HIV diagnosed late. The pre-diagnosis mean shows the mean outcome in the year prior to diagnosis for people with HIV. For each outcome, the effect size is defined as the magnitude of the point estimate of the treatment effect divided by the pre-diagnosis mean of that outcome for people with HIV and multiplied by one hundred. The number of observations is defined as the number of individuals observed in the year prior to diagnosis. The number of observations is lower for income and work hours due to differences in data availability in the source files. Robust and asymptotic standard errors are reported in parentheses. The stars represent the results of a standard two-sided t-test for whether the point estimates are statistically different from zero. Statistical significance is denoted as: * $p < 0.10$, ** $p < 0.05$, *** $p < 0.01$, where $p$ represents the p-value resulting from this test.

### Disability insurance receipt

Disability insurance (DI) in the Netherlands is administered by the Employee Insurance Agency (UWV), which determines the degree of disability of applicants based on their health status. During the first two years of illness, employers are legally obligated to continue paying individuals a minimum of 70% of their previous salary. Individuals who are unemployed or on temporary contracts can receive a Sickness Benefit in this period (Ziektewetuitkering). After two years of illness, the UWV assesses the individual's degree of disability based on their current and expected earning capacity. Individuals can enter one of two schemes: the WGA (Werkhervatting Gedeeltelijk Arbeidsgeschikten) for partially disabled individuals (35% - 80% degree of disability) or the IVA for fully disabled individuals (80–100% and permanent disability). Partially disabled individuals are expected to return to work, while receiving partial benefits, whereas fully disabled individuals are

not. Self-employed individuals do not qualify for this scheme and instead must rely on private, voluntary insurance. The Dutch DI system is considered relatively generous by the OECD, although, in contrast with other European countries, it places a strong emphasis on reintegrating individuals with a disability into the labor force[18]. Generally, having an HIV diagnosis is not in itself a sufficient condition to qualify for disability insurance, with worsening mental health conditions being a more likely reason for sick leave and disability claims. However, severe cases of AIDS-defining conditions can lead to work disability, which might explain the results described below.

Figure 4a shows the evolution of disability insurance receipt (defined as receiving disability benefits or sickness benefits for at least one month in a year) for people with HIV and matched controls. The groups have similar levels of disability insurance (DI) utilization pre-diagnosis, with 7.8 percent for people with HIV and 7.3 percent for the matched controls in the year prior to diagnosis. DI utilization increases to 11 percent in the year of diagnosis for people with HIV. Fig. 4b shows that this increase is driven primarily by the late-diagnosed group, who experience an increase in DI receipt from 8 percent in the pre-period to 12.8 percent in the year of diagnosis. Individuals not diagnosed late also experience an increase in DI receipt, from 7.6 percent before diagnosis to 9.7 percent in the year of diagnosis.

Figure 4c shows that an HIV diagnosis leads to a persistent increase in the receipt of DI. Table 2 reports that DI receipt increases with 46 percent after diagnosis for people with HIV. Fig. 4d splits the sample for the late- and non-late diagnosed group. Point estimates are zero in the pre-period but increase after diagnosis, which implies that people with HIV are more likely to receive disability or sickness benefits after diagnosis. Table 2 shows that DI receipt increases by 2.8 percentage points for the non-late diagnosed group, an increase of 36.8 percent, and by 4.9 percentage points for the late-diagnosed group, an increase of 60.5 percent.

### Robustness checks

One concern of our study design is selective attrition, i.e., individuals non-randomly leaving the sample in the post-HIV diagnosis period. In the setting of this paper, attrition can occur because of mortality or migration. In the study population of 5960, 319 people with HIV died after diagnosis and before the end of the sample period (5.3 percent) and 327 people with HIV migrated at least once after diagnosis (5.5 percent). In comparison, 1389 of the 59,600 matched controls died before the end of the panel (2.3 percent) and 2357 had at least one migration spell (4 percent). We first test for the impact of attrition by replacing missing post-diagnosis outcomes with the worst-case outcomes, i.e. one for disability insurance receipt and zeros for all other outcome variables. Column (2) of Supplementary Table SI-1 summarizes the average treatment effect on the treated. The magnitude of the point estimates increases slightly. If anything, selective attrition biases result towards zero. Given that this first test should be considered the worst-case scenario, we replace missing outcomes with the last observed value in column (3). The point estimates are similar to the main analysis. Attrition bias can also occur because for individuals diagnosed after 2015, we do not observe the full 7 years post-diagnosis. Therefore, as a third test, we restrict the sample to the cohorts 2010–2015 (3720 people with HIV compared to 5960 in the baseline analysis) that can be observed across all 7 years pre- and post-diagnosis. Column (4) shows that estimates are similar to the main results when restricting the analyses to these cohorts. Finally, we conduct a fourth test, which consists of excluding from the sample individuals who pass away before the final year of the panel, 2022. This removes 319 people with HIV and 1389 matched controls. Column (5) of Supplementary Table SI-1 shows that the estimated treatment effects are almost unchanged.

Even though the matched controls are randomly selected from the general Dutch population and matched to people with HIV on birth

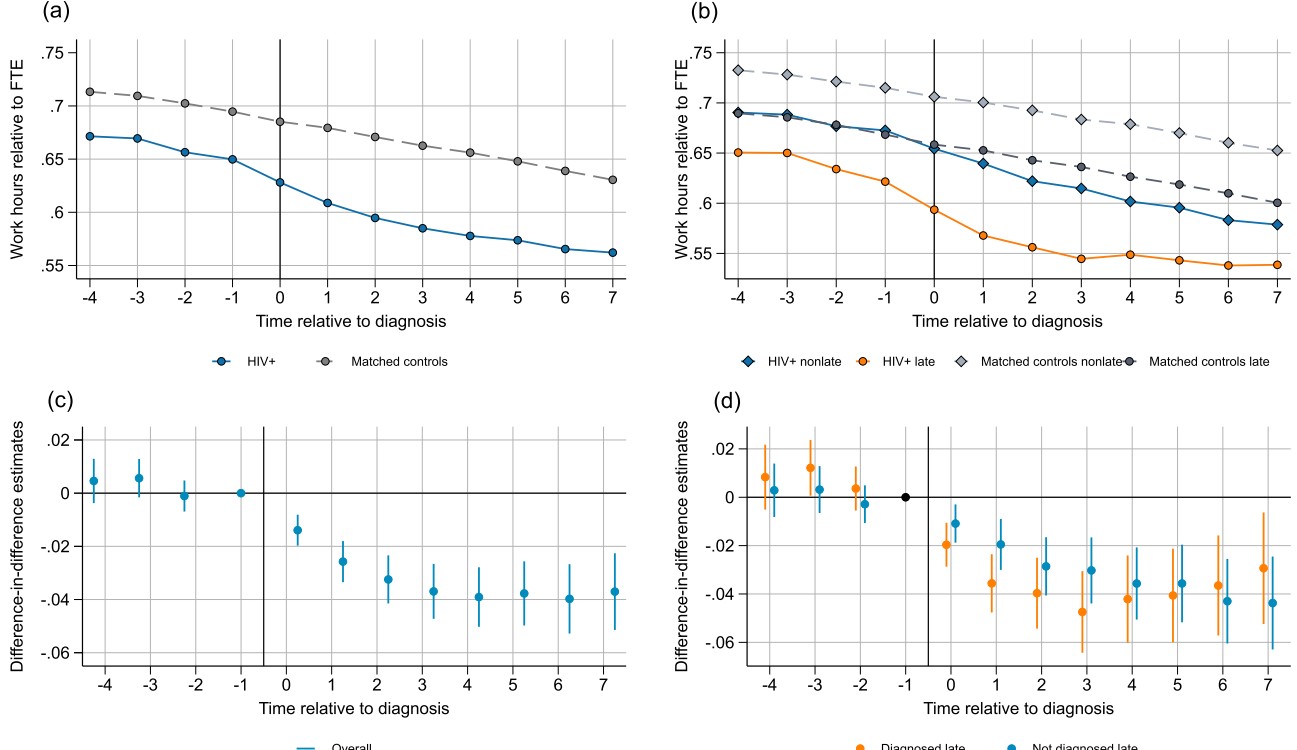

**Fig. 2 | Work hours relative to full-time employed.** Hours worked is defined in full-time equivalents (FTE), which implies that a value of one represents a full-time job, and zero that the individual does not work. Panel (**a**) People with HIV and matched controls shows the average hours worked over time relative to diagnosis (at t = 0) for people with HIV (solid blue line) and matched controls (dashed gray line). Panel (**b**) People with HIV and controls by disease stage makes an additional distinction between people with HIV and matched controls depending on stage at diagnosis: non-late (solid blue line and dashed light gray line, respectively) and late (solid orange line and dark gray dashed line, respectively). Panels (**c**) Difference-in-differences estimates and (**d**) Difference-in-differences estimates by stage report

event studies estimated using the Callaway and Sant'Anna estimator[50]. Each dot represents a point estimate for a specific period relative to diagnosis, and the bars represent the 95% confidence intervals, where t = 0 represents the year of diagnosis. In panel (**c**), N = 61,974 observations at period t-1. In panel (**d**), N = 33,792 observations at period t-1 for individuals not diagnosed late and N = 24,937 observations at period t-1 for individuals diagnosed late. The period t = −1 is the reference period and is standardized to zero. Data on hours worked is only available starting in 2006, which restricts the number of years before diagnosis that we can observe to four. Source data are provided as a Source Data file.

year, registered sex, education level and migration background, panels a and b of Figs. 1–4 showed that there exist differences in characteristics between people with HIV and matched controls prior to diagnosis. This may be explained by unobserved differences in relationship or health status. We perform two analyses to check the robustness of the results to using different control groups. First, we use individuals who have not yet been diagnosed with HIV as controls for earlier cohorts. The intuition is that individuals who receive an HIV diagnosis in a later period are arguably more similar to people with HIV than the general Dutch population on (un)observable characteristics. A drawback of this approach is lower statistical power, due to the smaller number of observations. Column (6) of Supplementary Table SI-1 shows that the results on income and DI are similar to the baseline results, but that the effect on employment is larger in magnitude. Second, we check the sensitivity of the results to randomly drawing five different samples of matched controls from the general Dutch population. Supplementary Fig. SI-1 shows that the point estimates are not statistically different across the five permutations, which implies that the results are not driven by the initial selection of the matched control group.

To flexibly account for potential violations of the parallel trends assumption, we apply the honest difference-in-differences method proposed by Rambachan and Roth (2023)[19]. This method allows us to bound our estimated treatment effects under violations of parallel trends of different magnitudes, for all outcome variables. Panels a – d of Supplementary Fig. SI-2 in the Supplementary Information show

violations of parallel trends of various relative magnitudes and the corresponding bounds placed on the estimated treatment effects. We highlight in red the "breakdown" magnitude, which is the size of the violation at which we can no longer reject the null hypothesis that the treatment effect is equal to zero. As shown in the figures, these breakdown magnitudes are relatively large. For instance, the parallel trends violation for income can reach a magnitude of 1.5 times the maximum pre-diagnosis drift before our estimates become statistically insignificant. In other words, if the violation of parallel trends in the post-diagnosis period is no more than 1.5 times the maximum pre-diagnosis violation, our results remain statistically significant. For employment, work hours and disability insurance receipt, these magnitudes are similar, ranging from 1 to more than 1.5, which implies that our results are robust and that even a relatively large violation of parallel trends does not invalidate them.

## Heterogeneity and mechanisms

To study the heterogeneity of our results, we repeat our descriptive analysis (panel a of Figs. 1–4), disaggregating the sample by age at diagnosis, migration background, and more precisely defined groups of HIV disease stage at diagnosis. Event studies are not estimated for these smaller subsamples because of concerns about too low statistical power. First, the sample is split between individuals aged below and above 45 years at diagnosis in Supplementary Figs. SI-2–SI-5. This cut-off represents an approximation of both the mean age at diagnosis in our sample, and of the middle-point of an individual's career.

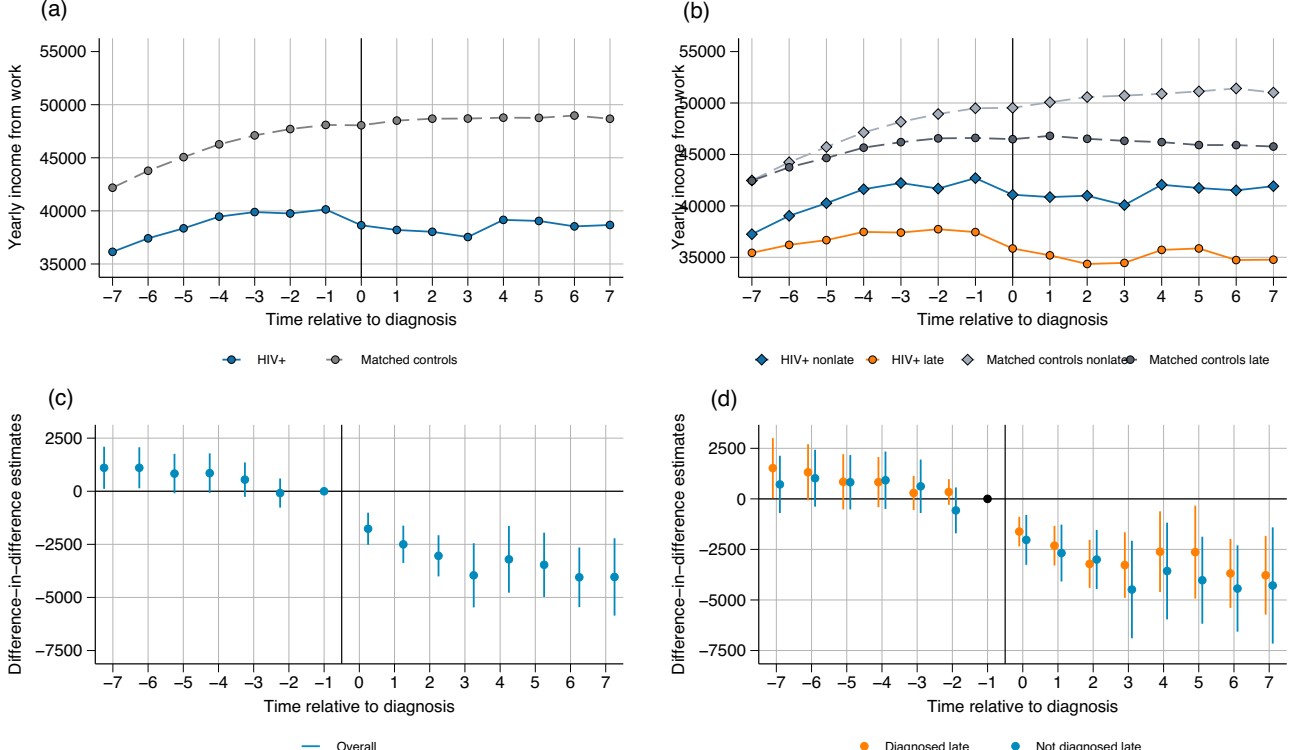

**Fig. 3 | Income from work.** Income is defined as gross annual income from work and self-employment. Panel (**a**) People with HIV and matched controls shows the average yearly income over time relative to diagnosis (at t = 0) for people with HIV (solid blue line) and matched controls (dashed gray line). Panel (**b**) People with HIV and controls by disease stage makes an additional distinction between people with HIV and matched controls depending on stage of diagnosis: non-late (solid blue line and dashed light gray line, respectively) and late (solid orange line and dark gray dashed line, respectively). Panels (**c**) Difference-in-differences estimates and (**d**)

Difference-in-differences estimates by stage report event studies estimated using the Callaway and Sant'Anna estimator[50]. Each dot represents a point estimate for a specific period relative to diagnosis, and the bars represent the 95% confidence intervals, where t = 0 represents the year of diagnosis. In panel (**c**), N = 64,416 observations at period t-1. In panel (**d**), N = 35,101 observations at period t-1 for individuals not diagnosed late and N = 25,916 observations at period t-1 for individuals diagnosed late. The period t = −1 is the reference period and is standardized to zero. Source data are provided as a Source Data file.

Post-diagnosis, the employment, work hours and income losses, as well as the increase in DI receipt experienced by people with HIV in both age groups are similar to those shown in Figs. 1–4. Second, the sample is stratified by migration origin in Supplementary Figs. SI-6–SI-9. Individuals are native Dutch if they and both of their parents are born in the Netherlands; they have a migration background if they or at least one of their parents is born outside of the Netherlands. Individuals with a Dutch, European or North American background experience smaller, but not negligible, losses in labor market outcomes post-diagnosis, compared to individuals with a Surinamese, South-American, Caribbean or Sub-Saharan African background. Individuals with a Middle Eastern, North African or Asian background also experience large losses, despite performing better in the labor market than their matched controls in the pre-diagnosis period.

Next, we stratify the sample using five ordinal groups of HIV disease stages at diagnosis in Supplementary Figs. SI-10–SI-13. The five groups in increasing disease severity are: individuals with a negative HIV test in the 12 months prior to diagnosis and/or diagnosis during an acute HIV infection, individuals with a CD4 count at diagnosis above 350 cells/mm³, individuals with a CD4 count between 200 and 350 cells/mm³, individuals with a CD4 count below 200 cells/mm³ and individuals with AIDS-defining illnesses at the time of diagnosis. These results confirm that all people with HIV experience a reduction in employment, work hours and income, and an increase in DI utilization post-diagnosis, regardless of HIV disease stage at diagnosis. However, similar to what we show in our main results, individuals with an early diagnosis and those with a CD4 cell count higher than 350 cells/mm³ at

diagnosis experience much smaller losses than individuals who are diagnosed in more advanced stages of an HIV infection.

We also conduct a heterogeneity analysis by recency of diagnosis, based on two groups, namely individuals diagnosed between 2010 and 2015, and individuals diagnosed between 2015 and 2022. This distinction highlights continual improvements in treatment and in the management of psychosocial aspects of HIV. We note that despite the observed differences in trends between the two periods observed across all four outcomes, the general patterns described in the main results remain. In all outcomes, there is a large drop in employment, work hours and income, and an increase in disability insurance utilization, after HIV diagnosis, among individuals diagnosed both before and after 2015. This concordance likely reflects similar approaches in the management of HIV between these periods.

Finally, we stratify the sample by registered gender, which could reflect differences in, for instance, childcare responsibilities, reproductive choices, as well as underlying differences in labor market outcomes. Across all outcomes, a clear pattern emerges: women experience worse labor market outcomes after an HIV diagnosis than men, perhaps reflecting differences in caring responsibilities or different levels of labor market attachment. These discrepancies are also likely explained by a difference in the prevalence of late-stage diagnoses in the female group, considering that the share of women with a late-stage diagnosis is 51.3%, compared to 41.3% among men.

To study potential mechanisms through which an HIV diagnosis might impact labor market outcomes, we examine individuals' mental healthcare expenditures, as identified in the data by basic health insurance reimbursements, before and after HIV diagnosis. All

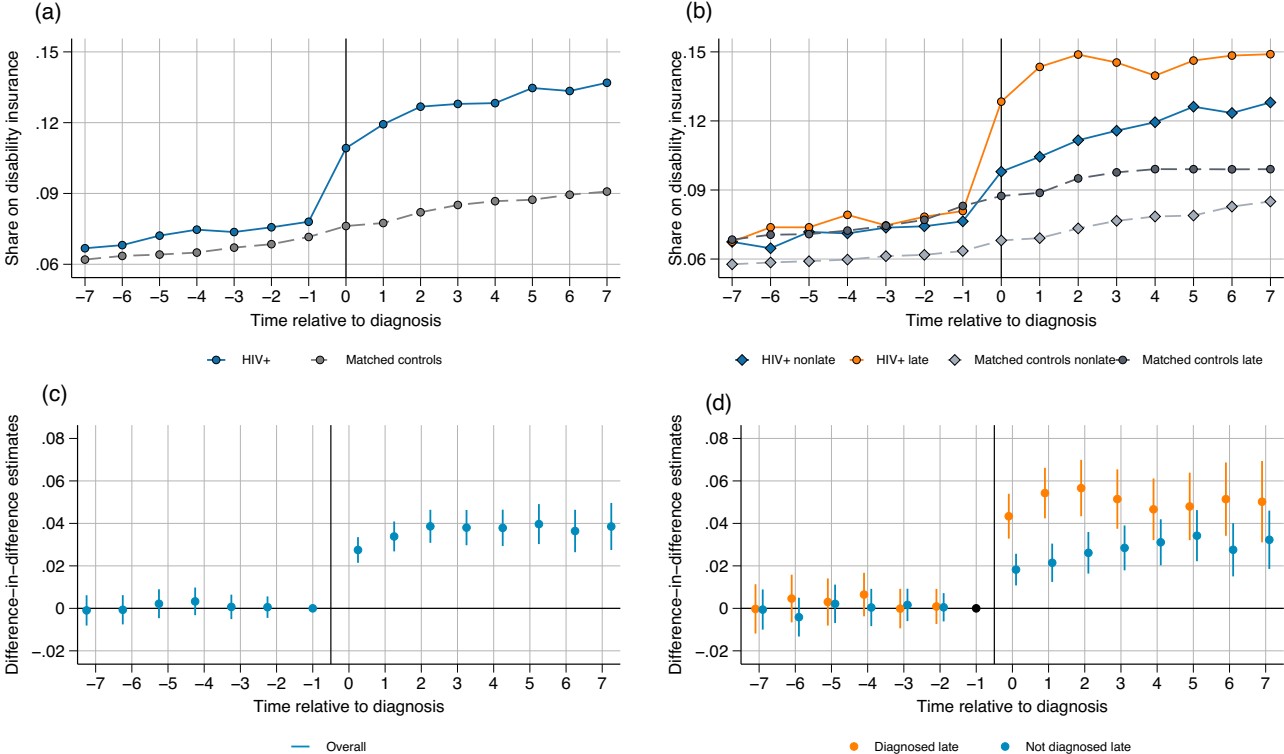

**Fig. 4 | Disability Insurance receipt.** Disability insurance (DI) take-up is defined as receiving disability or sickness benefits for at least one month in a given year. Panel **(a)** People with HIV and matched controls shows the proportion of individuals on DI over time relative to diagnosis for people with HIV (solid blue line) and matched controls (dashed gray line). Panel **(b)** People with HIV and controls by disease stage makes a distinction between people with HIV and matched controls depending on stage at diagnosis: non-late (solid blue line and dashed light gray line, respectively) and late (solid orange line and dark gray dashed line, respectively). Panels **(c)** Difference-in-differences estimates and **(d)** Difference-in-differences estimates by stage report event studies estimated using the Callaway and Sant'Anna estimator[50]. Each dot represents a point estimate for a specific period relative to diagnosis, and the bars represent the 95% confidence intervals, where $t = 0$ represents the year of diagnosis. In panel **(c)**, $N = 65,560$ observations at period t-1. In panel **(d)**, $N = 35,706$ observations at period t-1 for individuals not diagnosed late and N = 26,389 observations at period t-1 for individuals diagnosed late. The period $t = -1$ is the reference period and is standardized to zero. Source data are provided as a Source Data file.

individuals in the Netherlands are required by law to take up basic health insurance, which covers mental healthcare, so we expect that underreporting is not a concern. Supplementary Figs. SI–23 shows mental healthcare expenditures as reimbursed through the basic health insurance system before and after an HIV diagnosis. Panels a and b show the share of individuals with any mental healthcare costs (the extensive margin). These figures display strong parallel trends in the pre-diagnosis period, indicating that the share of people with HIV utilizing mental healthcare evolves at the same rate as the matched controls. People with HIV experience an increase in the probability of seeking mental health care after HIV diagnosis, whereas this is not observed for the matched controls. In panels c and d, we examine annual expenditures for mental health care (the intensive margin). These figures also show relatively flat parallel trends. Note that outliers, i.e., individuals with mental healthcare expenditures above the 97.5th percentile in any year in the pre-diagnosis period, are left out (183 dropped individuals, 0.4% of the sample with non-missing healthcare costs at t-1) because they have a large effect on consumption trajectories. While both groups experience an increase at the time of diagnosis due to only removing pre-diagnosis outliers, the discontinuity is much more pronounced for the people with HIV. These results provide evidence of a sharp rise in mental healthcare utilization around the time of diagnosis, suggesting that a deterioration in mental health is a potential mechanism for the effects of HIV on labor market outcomes.

Finally, access to disability insurance (DI) represents an important institutional feature of the Dutch welfare system and labor market. In

the Netherlands, individuals remain formally employed for the first two years of illness and continue being paid by their employers, after which their earnings capacity is assessed, and they either enter a scheme for fully and permanently disabled individuals or one for partly disabled individuals with an emphasis on reintegration into the labor market. While a majority (78.8%) of the individuals with HIV in our sample who are receiving DI benefits two years after diagnosis have a degree of disability of at least 80% (fully and permanently disabled) a relatively large share of 21.2% have lower degrees of disability that correspond to partial disability (12.9% have a degree of disability between 45% and 80%, and 8.4% have a degree of disability of less than 45%). Furthermore, 36% of individuals with HIV receiving DI or sickness benefits two years after diagnosis continue to be employed for at least one month, and a relatively small share of 23.4% of unemployed individuals are receiving benefits. Together, these numbers suggest that, while disability insurance utilization may explain a part of the observed effect of an HIV diagnosis on income and employment, these effects are not entirely compensated by an increase in DI utilization.

## Discussion
In this paper, we show that despite the universal accessibility of ART in the Netherlands, there is a significant negative impact of an HIV diagnosis on socioeconomic outcomes compared to a sample of matched controls. After an HIV diagnosis, individuals are 3.9 percent less likely to be employed, work 5.4 percent less FTE hours, have 8.9 percent lower annual incomes, and a 46.2 percent higher take-up of disability insurance (DI), compared to their pre-diagnosis outcomes.

Even in a country with universal healthcare, such as the Netherlands, health inequality remains an important issue. In the context of HIV, this can translate into a high share of individuals being diagnosed in a late or advanced stage. This can occur due to both psychosocial causes and (perceived or actual) barriers to healthcare services, with important factors including having a migration background, being a heterosexual male, and having a low income[16,20]. A qualitative study conducted in the Amsterdam region has identified psychosocial factors, such as low perceived risk or perceived stigma, and health system factors, such as perceived limited accessibility to HIV testing facilities, as reasons for avoiding HIV testing, and thus as risk factors for a late diagnosis[21]. Therefore, we also study whether the effects of receiving an HIV diagnosis vary by the disease stage at the time of diagnosis. For individuals who are diagnosed with late-stage HIV, we find a 6.3 percent decrease in employment, a 6.3 percent decline in FTE hours, 8.6 percent lower annual incomes, and a 60.5 percent higher utilization of DI. For the non-late diagnosed group, a group that is less likely to experience clinically relevant symptoms at diagnosis, but which might experience a gradual deterioration of somatic health, we still find a 2.6 percent decrease in employment, a 5.1 percent decrease in FTE hours, 9.2 percent lower annual incomes, and a 36.8 percent higher take-up of DI.

The negative effects for the late-diagnosed group in this paper are comparable to income and earnings losses that are observed after diagnosis with types of cancer with high survival rates (e.g., thyroid cancer, prostate cancer or breast cancer), such as a 9-10% reduction in income and a 2–4% reduction in the employment probability in the years after diagnosis[22]. Similarly, our results point in a similar direction to those identified in previous studies of the employment and work hour effects of other chronic conditions, such as hypertension, high blood cholesterol, and diabetes in Australia and several European countries[23,24]. We observe that after a late HIV diagnosis, some individuals exit employment, which is compensated with a higher utilization of DI. A late HIV diagnosis can affect labor market participation in multiple ways. Past research has shown that, among people with HIV, health status is an important determinant of employment[25–28]. Furthermore, individuals diagnosed late may experience changes in the incentives to work resulting from a lower utility from consumption[29], lower investments in human capital[30,31], lower preferences for market work[10,11], and may sort into less demanding, lower-paying occupations[32]. It is important to mention that the individuals included in our analyses were diagnosed in an era where ART side-effects were generally relatively minor compared to prior to 2010, as evidenced by an increasing number of studies[4,33–35], as well as clinical treatment guidelines[36]. This implies that any negative effects on socioeconomic outcomes are unlikely to be driven by adverse side-effects of ongoing antiretroviral treatment.

The significant negative estimates for the people who were diagnosed timely suggest that a deterioration in health is not the only mechanism through which HIV impacts an individual's labor market outcomes. The onset of chronic conditions, in general[37] and particularly in the setting of HIV[5–7] may be associated with a deterioration of mental health, which in turn can affect employment and welfare utilization. Stigmatization of and discrimination against people with HIV, in- and outside of the workplace, may also impact mental health, treatment adherence and economic outcomes[9,38,39]. The previously mentioned mechanisms related to labor market incentives and occupational preferences may also apply to timely diagnosed individuals, who may opt for a different labor-leisure trade-off after receiving an HIV diagnosis, regardless of experiencing symptoms at the time of diagnosis. Further work is needed to examine the exact mechanisms behind the labor market effects for both the late and the non-late diagnosed group.

Past research in low-income countries has shown that HIV has a negative causal impact on employment of 4–7 percentage points[40–42],

but also that a timely initiation of ART can mitigate part of this effect and lead to other positive outcomes[40,43–47]. Our results are in line with this literature, though somewhat smaller in magnitude, likely because many of these studies cover a period during which ART was less effective and presented a higher risk of adverse effects.

Our rich clinical dataset, based on a nationwide observational cohort of virtually all people in HIV care in the Netherlands, linked to administrative records from Statistics Netherlands, represents the key strength of this study. These data allow us to overcome the challenges of past studies investigating the socioeconomic effects of HIV in other high-income countries. These studies often use survey data, have small sample sizes and may suffer from selective non-response and attrition[13,25–28]. Two Danish studies have used administrative data to study the labor market impact of a medical innovation in antiretroviral therapy (HAART) in the 1990s. They find that access to ART improved labor market outcomes for individuals with HIV[14,48]. Another paper uses Swedish data to show that people with HIV are less likely to be employed than people without HIV[15]. Our study adds to the literature by estimating the causal impact of an HIV diagnosis on a variety of socioeconomic outcomes in a period and setting where ART is universally available and free of charge, and by making a distinction by HIV disease stage at the time of diagnosis.

Our study has two important limitations. First, we are unable to control for unobserved time-varying characteristics of people with HIV and their matched controls. This might explain why we see a small deviation in the pre-diagnosis trends in income between these two groups, raising some concerns about the comparability of our matched control group, for example on sexual behavior and/or orientation. Due to data limitations, we cannot match individuals on these characteristics, even though they could influence employment and lifestyle choices, which, in turn, can impact labor market outcomes. Despite this, we provide evidence that the parallel trends assumption holds in Figs. 1–4. Second, the generalizability of our results might be limited to other high-income countries. The Netherlands provides a "best-case" scenario in which to measure the effects of HIV, as HIV care is widely available, including for individuals without access to health insurance. In addition, disability insurance is mandatory for wage workers. Therefore, it is difficult to extrapolate these results to low- and middle-income settings, or to settings with less generous welfare systems.

In summary, this study shows that receiving an HIV diagnosis can impact individual outcomes beyond mortality and morbidity, and has long-term negative impacts on labor market outcomes regardless of the disease stage at the time of diagnosis. The negative impacts are larger for individuals who are diagnosed late. These findings hold in a setting with universal access to healthcare. Socioeconomic consequences of HIV may be even larger in settings with fewer resources or larger health care inequalities. If governments strive to lower the socioeconomic impact of HIV, it is key to understand the importance of efforts aimed at prevention and early detection of HIV. The findings in this paper also pave the way for interventions aimed at preventing individuals from dropping out of the labor market after receiving an HIV diagnosis.

## Methods
### Study design and data collection
HIV care in the Netherlands is provided by 23 designated treatment centers. The HIV Monitoring Foundation (Stichting hiv monitoring, SHM) is tasked by the Dutch Ministry of Healthcare, Welfare and Sports to monitor and report on all aspects of HIV-care for people with HIV in the Netherlands. Data collection was initiated in 1998 and data are prospectively collected in the ATHENA (AIDS Therapy Evaluation in the Netherlands) cohort, which represents over 98 percent of all people with HIV in care in the Netherlands[49]. People entering HIV care receive written material about participation in the ATHENA cohort, after which

they are asked to consent verbally to the use of their routinely collected medical data for research and monitoring (i.e., an "opt-in" procedure). Participants can withdraw their consent at any time. Data collection was approved by the ethical boards of all participating centers. Only routinely collected data were used for this analysis and therefore no additional review or consent was required.

Statistics Netherlands (Centraal Bureau voor de Statistiek, CBS) is an independent organization that collects, processes, and publishes reliable statistical data on residents of the Netherlands. The Statistics Netherlands Act constitutes the legal basis for Statistics Netherlands, and Statistics Netherlands is adherent to the European Union's General Data Protection Regulation. For the present study, data from the ATHENA cohort were linked to data from Statistics Netherlands using a probabilistic approach based on individual date of birth, postal code of last known residence, and sex at birth.

### Study population
A total of 28,294 people linked to HIV care in the Netherlands were successfully linked to data from Statistics Netherlands. The sample is restricted to 21,117 individuals aged from 28 to 62 at diagnosis (at least 3–5 years before the official retirement age in the studied period), ensuring that they are of working age and likely to be attached to the labor market. We study outcomes seven years pre-diagnosis, which implies that the youngest cohorts are at least 21 years old at the time of diagnosis. Individuals who received an HIV diagnosis outside of the Netherlands are removed from the data as pre-diagnosis outcomes cannot be observed for this group. The sample is further restricted to 7113 individuals who were diagnosed between 2010 and 2022, as the labor market data starts in 2003 and we cannot observe seven years of pre-diagnosis data for individuals diagnosed before 2010. Focusing on the 2010–2022 cohorts also ensures that the study population was likely not treated with more toxic and less efficacious treatment regimens and did not experience prolonged deferrence of ART initiation[4,33].

The sample is further restricted to 5960 individuals whose employment and disability insurance receipt data is available for the entire period of seven years before diagnosis. Individuals who are removed from the sample in this way are younger than those who remain (37.8 years versus 43.6 years), less likely to be male (73.7% versus 87.6%) and more likely to have a first generation migration background (86.6% versus 23.9%). This selection step is done in order to maintain a balanced panel in the pre-diagnosis period. We impute missing incomes for 252 individuals with at most 2 periods of missing income data, which leaves 114 individuals for whom income cannot be imputed. There are 326 individuals with HIV whose work hours are not observed at any point in the pre-diagnosis period, or whose matched controls' works hours are not observed in the pre-diagnosis period. Individuals whose income and work hours are not observed for the entire pre-diagnosis period do not enter the model for the respective outcome.

The administrative data are used to select a control group with characteristics similar to the sample of people with HIV. We start with 17,651,946 million individuals who were ever registered in the Netherlands since 1995 and whose employment, DI receipt and income are observed for at least seven consecutive years at any point starting in 2003. Individuals who are observed in the clinical data are excluded from this group. We conduct 10–to–1 exact matching, which implies that each person with HIV is matched with 10 random eligible matches who have the same birth year, registered sex, migration background (native, first generation, and second generation) and highest education level achieved (low, medium, high, or missing). A potential match should be observed in the employment, DI and income data for seven consecutive years prior to the year of diagnosis of the matched person with HIV. This implies that labor market outcomes of matched controls are observed at the same ages as the matched person with HIV.

The final sample consists of 5960 people with HIV and their 59,600 matched controls for whom employment and disability insurance can be observed seven years pre-diagnosis. Of these, we observe seven years of income data pre-diagnosis for 5856 people with HIV and 58,560 controls, and four years of work hours data for 5634 people with HIV and 56,340 matched controls.

### Variables
The clinical data contain information on the date of diagnosis, ART initiation, and AIDS defining illnesses (e.g. Kaposi's sarcoma, tuberculosis, toxoplasmosis, pneumocystis pneumonia, candidiasis, wasting syndrome), CD4 cell count, plasma HIV-1 RNA viral load at the time of diagnosis and at subsequent measurements, most recent negative HIV test, and having achieved viral suppression. A late-stage diagnosis is defined as having a CD4 cell count below 350 cells/mm³ at diagnosis and not having a negative HIV test in the 12 months prior to diagnosis, or by having an AIDS-defining illness at diagnosis and no negative test in the previous 12 months. The not-late diagnosed group consists of individuals with a CD4 cell count above 350 cells/mm³ at diagnosis and no evidence of AIDS-defining illnesses, or those who have had a negative HIV test within the 12 months prior to diagnosis. This results in 3246 individuals with a non-late diagnosis, 2399 individuals with a late diagnosis, and 315 individuals who could not be assigned to either group due to missing data.

From Statistics Netherlands, we use information on year of birth, registered sex, migration background, and the highest level of education achieved. Information on employment, work hours and income is available from tax records, and disability insurance utilization is reported by the Employee Insurance Agency (Uitvoeringsinstituut Werknemersverzekeringen, UWV). An individual is classified as employed in a given year if they had an employment contract for at least one month in that year, and an individual is classified as receiving disability insurance if they receive benefits for at least one month in that year. Work hours are in full-time equivalents (FTE), meaning that a value of 1 indicates having a full-time job and 0 indicates being unemployed. Disability and sickness benefits include benefits for individuals who are unemployed or on temporary contracts, as well as disability insurance benefits that start after two years of sickness leave for employed individuals. Information on income from employment is corrected for inflation. All data are available until 2022, which implies that we do not observe a complete seven-year follow-up period after diagnosis for individuals diagnosed in 2016 or later. We show in a robustness check that restricting the analysis to the cohorts 2010–2015 does not change the results.

### Empirical design
To estimate the dynamic effects of an HIV diagnosis on individuals' labor market outcomes, we apply the doubly robust staggered difference-in-differences estimator as developed by Callaway and Sant'Anna[50]. This is done using the *csdid* command in Stata[51]. The model is an extension of the canonical dynamic Two-Way Fixed Effects model as shown in Eq. (1):

$$Y_{it} = \sum_{k=7, k \neq -1}^{7} \beta_k \mathbb{1}\left[g + k = t\right] + \alpha_i + \lambda_t + \gamma X_{it} + \in_{it} \tag{1}$$

where $Y_{it}$ is a labor market outcome for individual $i$ in calendar year $t$. Indicator variable $\mathbb{1}[g + k = t]$ is equal to one for each year $k$ relative to the diagnosis year $g$, the parameter $\beta_k$ captures the evolution of labor market outcomes of people with HIV compared to the matched control group relative to the year prior to diagnosis ($k = -1$). $\alpha_i$ are individual fixed effects that capture time-invariant characteristics (e.g., a person's risk attitudes), $\lambda_t$ are time fixed effects that capture time-varying factors (e.g., year-specific economic shocks), $X_{it}$ is a vector of control variables that includes registered sex, age, education level and

migration background, and $\varepsilon_{it}$ are robust and asymptotic standard errors.

Two-way fixed effects models are shown to provide biased estimates in cases of treatment effect heterogeneity which is likely present in cases with staggered treatment[52]. This is likely the case in our context, as the treatment is the year of diagnosis, which occurs in different calendar years for different individuals. Therefore, Eq. (1) is estimated using the Callaway and Sant'Anna estimator[50]. The main parameters estimated by the model are the group-time average treatment effects on the treated $ATT(g,t)$, which represent the effect in calendar year $t$ for people diagnosed in calendar year $g$. These are aggregated into weighted averages across all cohorts and years on a scale that measures time relative to diagnosis $(t–g)$. Panels c and d of Figs. 1–4 show event-study figures in which these aggregates are plotted over time relative to diagnosis using the Callaway-Sant'Anna estimator with a 14-year window around diagnosis. The $ATT(g,t)$ estimates are also aggregated into a before-after estimator in Table 2 which summarizes the effect of an HIV diagnosis.

Two assumptions are needed to interpret the estimates as the causal effect of an HIV diagnosis. First, labor market outcomes of people with HIV should not adjust in anticipation of the HIV diagnosis. Panels c and d of Figs. 1–4 show that the deviation from the pre-diagnosis trend only starts in the year after diagnosis for all outcome variables considered. This suggests that individuals do not anticipate an HIV diagnosis. Second, labor market outcomes of people with HIV should have followed the same trend as the matched controls in the absence of an HIV diagnosis (i.e., the parallel trends assumption). Panels c and d of Figs. 1–4 show that the labor market outcomes of people with HIV and the matched controls develop in parallel in the years prior to diagnosis and that there are no statistically significant deviations in these pre-trends. Hence, this suggests that the parallel trends assumption holds.

The results presented are based on calculations carried out by the Stichting HIV Monitoring (SHM) in project number 8944 using non-public microdata from Statistics Netherlands (CBS) and Vektis CV. Analyses were conducted using STATA (version 16.0, StataCorp, College Station, TX, USA).

### Ethics statement
The data collected from participants in the ATHENA cohort is part of routine monitoring. At initiation, the cohort was approved by the institutional review board of all participating centers. The participants provide consent to use ATHENA data for research purposes and also allow linkage to other data sources as Statistics Netherlands. (https://www.hiv-monitoring.nl/en/what-we-do/information-people-living-hiv/patient-information-sheet). Information for participants about data linkage is listed on the website of Stichting HIV Monitoring (https://www.hiv-monitoring.nl/en/research-using-our-data/datakoppelingen). Statistics Netherlands data is only accessible by authorized investigators. The usage and analysis of the combined dataset is allowed under the Dutch CBS ( = Statistics Netherlands) law (https://wetten.overheid.nl/BWBR0015926/2022-03-02/#Hoofdstuk5_Paragraaf1_Artikel33). Output is independently checked by Statistics Netherlands.

### Reporting summary
Further information on research design is available in the Nature Portfolio Reporting Summary linked to this article.

## Data availability
All results presented here are calculated from non-public registry data from Centraal Bureau voor de Statistiek (CBS), accessed through the Remote Access environment. CBS was not involved in the calculation of any of the results presented. While the data are not publicly available, academic institutions can apply for access to the Remote Access environment through the CBS (for additional information, see https://www.cbs.nl/en-gb/our-services/customised-services-microdata/microdata-conducting-your-own-research. ATHENA cohort data (without CBS data) used in this study are available upon reasonable request. Requests for data access can be made to: hiv.monitoring@amsterdamumc.nl. Requests will be reviewed on a case-by-case basis. Statistical information or data for separate research purposes from the ATHENA cohort can be requested by submitting a research proposal to SHM (https://www.hivmonitoring.nl/english/research/research-projects/). The proposal will undergo review by representatives of SHM for evaluation of scientific value, relevance of the study, design, and feasibility, statistical power, and overlap with existing projects. Source data are provided with this paper.

## Code availability
The code used for the analysis is based on existing commands within the Stata 16 software. No custom algorithms or software were developed for this analysis. The analysis is conducted within the restricted remote access microdata environment of Statistics Netherlands (Centraal Bureau voor de Statistiek, CBS). The code is available at: https://uvaauas.figshare.com/articles/software/Labor_Market_Outcomes_of_People_with_HIV_Pre-_and_Post-Diagnosis_in_the_Netherlands/30738545.

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

## Acknowledgements

The ATHENA Cohort (Supplementary Note 1) is managed by Stichting hiv monitoring and supported by a grant from the Dutch Ministry of Health, Welfare, and Sport through the Centre for Infectious Disease Control of the National Institute for Public Health and the Environment. The collaboration project [LSHM23014-SGF] is co-funded by the PPP Allowance made available by Health~Holland, Top Sector Life Sciences & Health, to stimulate public-private partnerships. We further acknowledge funding from the Amsterdam Dinner Foundation. We are grateful to Michiel Heidenrijk, Mette Ejrnaes, Job van Exel, Bruno Ventelou, Daniela Bezemer, Anders Boyd, Colette Smit, Mirjam Schulpen, seminar participants at the University of Amsterdam, Tinbergen Institute, conference participants at the CINCH-dggö Academy in Health Economics, the European Health Economics Association PhD Conference, lolaHESG, NCHIV 2024, and the U21 Health Sciences Group 2024 Annual Meeting for helpful comments and suggestions. We thank Statistics Netherlands and the Amsterdam Health Technology Institute for assisting with the use of the data. Generous financial

support from the Amsterdam Research Center for Health Economics (ARCHE) is gratefully acknowledged.

## Author contributions

M.v.d.V., M.P., W.J., A.T., and E.Z. conceptualized and designed this study. M.v.d.V., F.W., V.J., A.v.S., and A.T. were involved in the data management. A.T. and E.Z. were involved in data analysis. All authors were involved with interpretation of the data. A.T. and E.Z. drafted the manuscript, all other authors were involved in revising it. All authors read and approved the final manuscript.

## Competing interests

MvdV received unrestricted research grants and fees for participation in advisory boards from Gilead Sciences, MSD and ViiV, all paid to his institution. FW received fees for advisory boards from ViiV Healthcare. AvS reports grants, paid to his institution, from the Dutch Ministry of Health, Welfare and Sport through the Centre for Infectious Disease Control of the National Institute for Public Health and the Environment, and from the European Centre for Disease Prevention and Control (ECDC). All other authors declare no competing interests.
