## [Transparent Peer Review file · Nature Communications]

Labor Market Outcomes of People with HIV Pre- and Post-Diagnosis in the Netherlands

Corresponding Author: Mr Andrei Tuiu

Version 0:

Reviewer comments:

Reviewer #1

(Remarks to the Author)

Review comments

Summary

The aim of the paper is to estimate the causal impact of receiving a diagnosis of HIV on labor market outcome such as employment, working hours, labor market income and disability insurance receipt. The authors convincingly show that the HIV diagnosis has adverse effects on labor market outcomes using an event study design. The empirical analyses are competently executed and a series of robustness checks confirm that the results are robust. The authors also carefully explain why it is interesting to consider labor market effects of an HIV diagnosis in a Dutch context.

General comments

The paper concerns an important and interesting topic. The paper contributes to the existing literature on the impact of HIV on socio-economic outcomes.

The paper convincingly documents that the diagnosis of HIV leads to worse labor market outcomes, but is more brief on the mechanisms that are causing the deterioration of labor market outcomes. There are at least five possible mechanisms by which the diagnosis of HIV could affect labor market outcomes:

- a. Deterioration of somatic health
- b. Side effects of the ART treatment
- c. Deterioration of mental health
- d. Discrimination and stigma of HIV patients
- e. Eligibility for Disability Insurance (DI)

The paper discusses the first four explanations but does only provide limited empirical evidence of the mechanism. The authors rule out that the effect of the diagnosis is solely driven by deterioration of somatic health since they also find negative labor market effect for the “non-late diagnosed group” which is less likely to experience symptoms from HIV. An additional argument could be that the deterioration of somatic health is likely to happen gradually and not in particular around the time of diagnosis. Furthermore, they argue that the side-effects of ART treatment, which many HIV patients receive, are less severe since this study is based on relatively recent diagnosed patients and the side-effects are much milder now than they used to be. Given that the authors also have access to clinical data such as CD4 counts and related measures it would be interesting to see empirical evidence that could support that neither somatic health deterioration nor side-effects ART treatment explain the decline in labor market outcome around the time of diagnosis.

The authors interpret their results in the light of deterioration of mental health as well as HIV patients may be exposed to stigmatization and discrimination. These explanations seem plausible, but it would have been good to provide empirical support for these explanations, e.g. investigations of the usage of mental health services before and after the diagnosis. An additional mechanism that is not discussed in the paper is the eligibility of DI. The paper is silent about the institutional setting of DI in the Netherlands. If DI is only granted to individuals with a diagnosis of a chronic disease, it is unsurprising that the fraction of recipients goes up after the HIV diagnosis is given. The paper would benefit from more details on the Dutch DI system. Furthermore, this raises the question, if the impact seen in this study also would apply to other countries with a less or more generous DI system for HIV patients. Therefore, it would be useful to know how much of the observed decline in employment, income and working hours is driven by individuals applying and receiving DI after being diagnosed with HIV. Table 2 indicates that the decline in employment of about 2.8 percentage points for HIV patients can entirely be explained by the increase of 3.6 percentage points higher rate of DI recipients. Similarly, part of the decline in labor income could be due to DI. It would be useful to make a decomposition of employment effect to see how much of the decline can be explained by more DI recipients. If it turns out that the decline in labor market outcome can be explained by an increase in DI

recipients, it has important policy implications.

Specific comments

In the introduction it should be highlighted under which assumption that the staggered Difference-in-Difference estimation will identify the causal effect. In section 4.4 these assumptions are discussed but it should also be stated in the introduction. In figure 3, the pre-trend for group diagnosed late is not flat. This could be an indication of anticipation effects. It is also a bit surprising that the income effect is bigger for the non-late diagnosed group compared the late diagnosed group.

The distinction between late and non-late diagnosis is interesting and the analyses show bigger effects for the individuals with late diagnosis. It is also clear that there are systematic differences between the two groups, where the late diagnosis group contains a higher share of individuals with unknown education and 1st generation immigrants. The paper does not explain why some individuals are diagnosed late in a universal health care system with relatively low health inequality.

Reviewer #2

(Remarks to the Author)

This paper examined the impact of HIV diagnosis on socioeconomic outcomes in the Netherlands, including employment, disability benefits, and income. The authors linked clinical data from the Dutch ATHENA cohort of people with HIV with administrative (employment/income) data for the Dutch population, using a difference-in-difference design. They found that people with HIV were less likely to be employed, worked fewer hours, earned less income, and were more likely to receive disability benefits for up to 7 years after the HIV diagnosis. These effects were more pronounced for those diagnosed with late-stage disease. Those with non-late-stage disease also experienced a deterioration of socioeconomic outcomes, albeit less pronounced, demonstrating a dose/response relationship.

The study shows that there is a significant negative impact of an HIV diagnosis on socioeconomic outcomes compared to a sample of matched controls. These findings confirm previous observations, but with a large sample size, in a high-resource setting with universal accessibility of ART and using a rigorous methodology to study causation. The authors created a control cohort matched on sex, age, migration status and education level, and performed a number of sensitivity analyses to help establish causation. The paper is very well-written and the results support the conclusions. As such these findings can be of help to researchers and public policy practitioners who model the economic impact of HIV and the cost-effectiveness of prevention and early treatment programs for settings such as Netherlands or similar settings.

Comment:

There are differences observed with regards to work/income even prior to HIV diagnosis, which, as the authors also note, could be due to unmatched differences with regards to preferred types of employment or other social impacts for people in gender minorities. Given that the majority of HIV infections in high resourced settings occur among men who have sex with men, an additional sub-analysis for men might include a control group of men who have sex with men without HIV; or perhaps a group of men with another STI (perhaps a rectal STI). If such data sources are available, the authors might want to consider this comparison, as it would match more closely the sample of people with HIV (for men), strengthening the claim of causality and assessment of size of impact. Sources could be surveys or medical records, even though the sample size may not be as large.

Reviewer #3

(Remarks to the Author)

The investigators present a longitudinal matched analysis, comparing people with and without HIV in the Netherlands in terms of employment-related measures, and including the pre and post HIV-diagnosis periods. The powerful aspect of this study is the use of linkage of rich national databases that provide information on demographic, HIV and labour market variables, and allow the selection of a relevant control group. A thoughtful, appropriate and well explained analysis has been undertaken, that attempts to conduct a controlled comparison between people with and without HIV, as well as considering if the results are robust in a range of sensitivity analyses.

In the abstract I would state the restriction to the period 2010-2022 – I think this strengthens the analysis because the early period of the HIV epidemic was very different and inferences about employment based on that period will have much less relevance now.

In the introduction, the background of the question, the aims and broad methodology are well explained, but there is no statement of rationale for the work saying why it is important to address this question.

Is there a general methods paper for the ATHENA cohort that could be referenced for further details of cohort methodology and regulatory processes?

Some points regarding the results. Table 1 – why is there no missing data for migration status as there is for education? Table 2 – please explain the definition of the effect size in the footnote.

Relating to Figure 1 – please explain why a general downward trend is observed in employment with time – would this be expected with age, or does it reflect methodological aspects of the comparison? Also of interest to comment on (in

results/discussion) is the fact that when comparing people with and without HIV, the widening of the pre-diagnosis gap occurs from diagnosis to two years afterwards – following this the lines run parallel again, suggesting quite an immediate effect of HIV on employment after diagnosis, rather than a continually widening gap subsequently. In fact, graph (d) implies some ‘catch-up’ in the late diagnosis subgroup over time. Similar patterns are apparent for most of the other outcomes in the remaining figures – please give your interpretation of these patterns.

In addition, the contracted scales on the graphs perhaps overstate some differences a little –e.g. a few percentage points for employment on Figure 1, suggesting a relatively modest difference. However, differences do appear greater for some of the other employment measures. From this perspective, it would be useful to have some more information on how the magnitude of difference for HIV compares to that for other chronic conditions where analyses, outcomes and time period are broadly comparable – this is touched on in the discussion, it would be nice to see this expanded somewhat.

Although the investigators matched on a number of demographic factors (sex, age, migration status, education), they did not match on sexuality. I assume such data are not available, but this could be relevant, as people with HIV are more likely to be gay/bisexual men/MSM compared to the general population. This subgroup could have different employment profiles (leading to an under or over-estimate of HIV versus non-HIV differences). For example they are perhaps less likely to have caring responsibilities, which impact employment options and choices. They may also have elevated levels of some specific lifestyle factors such as smoking and drug use which may also act as confounders in the comparison. Although additional confounding is mentioned in the discussion, there is no consideration of specific factors, which I think is needed.

The section on heterogeneity is useful - I would also like to suggest two further stratifications. First, based on period of diagnosis, because even in this restricted 12-year period (2010-2022) there have been continual improvements in treatment as well as potential changes in management of the psychosocial aspects of HIV. Looking at those with earlier and later diagnoses within this period would shed light on whether employment outcomes relative to those without HIV are improving for those diagnosed more recently. Second, related to my earlier comment - based on sex, because childcare responsibilities are likely to impact considerably on employment, and certain subgroups affected by HIV (MSM) may be less likely to have these responsibilities (see my previous comment) it would be useful to look at the analysis in the female subgroup particularly. Still on this theme, another issue is that having HIV may impact reproductive choices, which would impact employment patterns.

In the methods the selection processes for inclusion are outlined. 15-20% of the people diagnosed between 2010 and 2022 are excluded because employment and disability insurance receipt data are not available for the entire period of seven years before diagnosis. Is this group likely to be different in terms of employment or other factors than those with data? These potential biases should be considered in the discussion..

Version 1:

Reviewer comments:

Reviewer #1

(Remarks to the Author)

I have read the revised manuscript, and I think the authors have done a good job of revising the text such that it accommodates most of my previous comments. However, I have two remaining issues.

1. Mental Health Expenditures and Parallel Trends Assumption: I appreciate the inclusion of analyses on mental health expenditures in the current version. However, as noted by the authors in line 408, the trends in mental health care expenditures do not follow parallel trends. This is concerning, as it violates the assumption of parallel trends, which is fundamental to causal inference in event study designs. Consequently, the authors have chosen to present only descriptive evidence for mental health expenditures, which is a reasonable decision.

Nonetheless, this raises questions about the validity of the analyses for other outcome variables—namely employment, hours worked, income, and disability insurance (DI). If the HIV and control groups exhibit non-parallel pre-trends in mental health expenditures, it suggests that the groups may differ systematically even before diagnosis, with the HIV group potentially experiencing deteriorating mental health prior to diagnosis.

To address this concern, the authors should provide evidence that the estimated effects on labor market outcomes are not driven by these differential trends in mental health. One possible approach would be to construct a matched control group that exhibits similar trends in mental health expenditures. For example, matching on the level of mental health expenditures in periods -7 and -1 could help ensure comparability.

2. Disability Insurance (DI): The argument presented in lines 417–429 lacks clarity. It is not evident that the analyses can rule out the possibility that the observed effects on employment, hours worked, and income are driven by DI.

One way to provide suggestive evidence would be to replicate the analyses on a subsample of individuals who never receive DI. If significant effects on employment and income persist within this subsample, it would strengthen the argument that the observed outcomes are not solely attributable to DI.

Reviewer #3

(Remarks to the Author)

Version 2:

Reviewer comments:

Reviewer #1

(Remarks to the Author)

I have read the revised manuscript and the replies from the authors, and I think the authors have done a good job of revising the text such that it accommodates most of my previous comments. I do not have further issues.

Point by Point Responses to Comments from Referee 1

We sincerely thank the referee for their thorough and constructive comments on our paper, “Labor Market Outcomes of People with HIV Pre- and Post-Diagnosis”. We greatly appreciate the time and effort spent on providing feedback, which has been instrumental in improving the quality of our paper.

Below, we provide a point-by-point response to all your comments. In our responses, bold text represents your original comments, plain text provides our answers, and italicized text includes direct quotes from the revised manuscript. Note that we marked changes to the text in the manuscript in red.

The paper concerns an important and interesting topic. The paper contributes to the existing literature on the impact of HIV on socio-economic outcomes. The paper convincingly documents that the diagnosis of HIV leads to worse labor market outcomes, but is more brief on the mechanisms that are causing the deterioration of labor market outcomes. There are at least five possible mechanisms by which the diagnosis of HIV could affect labor market outcomes: (a) Deterioration of somatic health, (b) Side effects of the ART treatment, (c) Deterioration of mental health, (d) Discrimination and stigma of HIV patients, (e) Eligibility for Disability Insurance (DI)

The paper discusses the first four explanations but does only provide limited empirical evidence of the mechanism. The authors rule out that the effect of the diagnosis is solely driven by deterioration of somatic health since they also find negative labor market effect for the “non-late diagnosed group” which is less likely to experience symptoms from HIV. An additional argument could be that the deterioration of somatic health is likely to happen gradually and not in particular around the time of diagnosis.

We thank the reviewer for their helpful remarks. In the responses below we will address points (b), (c), (d) and (e) separately. Regarding point (a), we acknowledge the possibility that the deterioration of somatic health occurs gradually, and not necessarily as a shock around the time of the diagnosis. Therefore, the manuscript was adjusted as follows:

Old (lines 332 – 335): *For the non-late diagnosed group, a group that is less likely to experience clinically relevant symptoms at diagnosis, we still find a 2.6 percent decrease in employment,*

a 5.1 percent decrease in FTE hours, 9.2 percent lower annual incomes, and a 36.8 percent higher take-up of DI.

New (lines 446 – 450): *For the non-late diagnosed group, a group that is less likely to experience clinically relevant symptoms at diagnosis, **but which might experience a gradual deterioration of somatic health**, we still find a 2.6 percent decrease in employment, a 5.1 percent decrease in FTE hours, 9.2 percent lower annual incomes, and a 36.8 percent higher take-up of DI.*

Furthermore, they argue that the side-effects of ART treatment, which many HIV patients receive, are less severe since this study is based on relatively recent diagnosed patients and the side-effects are much milder now than they used to be. Given that the authors also have access to clinical data such as CD4 counts and related measures it would be interesting to see empirical evidence that could support that neither somatic health deterioration nor side-effects ART treatment explain the decline in labor market outcome around the time of diagnosis.

To support our assumption that a gradual deterioration of somatic health after HIV diagnosis related to ineffective ART and subsequent somatic symptoms are unlikely to explain our results, we would like to draw the reviewer’s attention to Table 1, which shows the proportion of individuals with a suppressed HIV-1 RNA one year after diagnosis by stage at diagnosis. The large proportion shown in Table 1 (more than 97%, regardless of the stage of HIV at diagnosis) clearly demonstrates that viral loads were successfully suppressed by the use of ART one year after diagnosis.

Unfortunately, it is not possible to report the impact of side-effects of treatment since these are only collected systematically in the ATHENA cohort for individuals who experience a change in treatment. However, we would like to highlight a recent body of literature¹⁻⁴, as well as HIV treatment guidelines that highlight a strong improvement in the tolerability and efficacy of ART over time⁵. While this evidence does not completely rule out the possibility of ART-related side-effects, we argue that, in the current era, severe adverse events that can impact labor market outcomes are unlikely to occur.

We make the following edits to the manuscript:

Old (lines 345 – 348): *It is important to mention that the individuals included in our analyses were diagnosed in an era where ART toxicity was much less frequent and severe than prior to 2010^{28,4}, which implies that any negative effects on socioeconomic outcomes are unlikely to be driven by adverse side-effects of treatment.*

New (lines 464 – 468): *It is important to mention that the individuals included in our analyses were diagnosed in an era where **ART side-effects were generally relatively minor compared to prior to 2010, as evidenced by an increasing number of studies¹⁻⁴, as well as clinical treatment guidelines⁵. This implies that any negative effects on socioeconomic outcomes are unlikely to be driven by adverse side-effects of **ongoing antiretroviral** treatment.***

The authors interpret their results in the light of deterioration of mental health as well as HIV patients may be exposed to stigmatization and discrimination. These explanations seem plausible, but it would have been good to provide empirical support for these explanations, e.g. investigations of the usage of mental health services before and after the diagnosis.

We thank the reviewer for their insightful comment regarding the relationship between HIV and mental health, stigmatization and discrimination. We add an additional analysis to the Supplementary Appendix (Figure SI-23, also included below), in which we show mean mental healthcare expenditures for people with HIV and the matched controls, both overall and by stage at diagnosis, before and after diagnosis. Even though we cannot directly observe stigmatization and discrimination in the administrative data, the analysis of mental health expenditures may include cases where experiences of stigmatization or discrimination contribute to mental health problems.

These expenditures include the costs of diagnosis-treatment combinations consisting of consultations and treatments. We find that people with HIV generally have higher mental health expenditures than the matched controls, both in general (Figure SI-23, panel a) and when separating the sample by stage of HIV at diagnosis (Figure SI-23, panel b). Interestingly, individuals diagnosed late follow a different trend compared to those not diagnosed late. Whereas those with a timely diagnosis experience a persistent increase in mental healthcare costs starting before diagnosis and continuing in the post-diagnosis period, individuals

diagnosed late experience a sharp increase in costs around the time of diagnosis, followed by a large decline.

We change the title of section 2.7 Heterogeneity to **2.7 Heterogeneity and mechanisms**. We also make the following edit to this section to introduce and discuss this new result:

New (lines 403 – 416): *To study potential mechanisms through which an HIV diagnosis might impact labor market outcomes, we examine individuals' mental healthcare expenditures, as identified in the data by basic health insurance reimbursements, before and after HIV diagnosis. All individuals in the Netherlands are required by law to take up basic health insurance, which covers mental healthcare, so we expect that underreporting is not a concern. We present this data as a descriptive analysis because of concerns about violations of the parallel trends assumption. Figure SI-23 shows mental healthcare expenditures as reimbursed through the basic health insurance system. We find that people with HIV generally have higher mental healthcare expenditures than the matched controls, both in general (Figure SI-23, panel a) and when separating the sample by stage of HIV at diagnosis (figure SI-23, panel b). Interestingly, individuals diagnosed late follow a different trend compared to those not diagnosed late in the pre-diagnosis period. However, whereas those with a timely diagnosis experience a persistent increase in mental healthcare costs which seems to accelerate post-diagnosis, individuals diagnosed late experience a sharp increase in costs around the time of diagnosis, followed by a large decline.*

Figure SI-23: Mental healthcare costs

Note: Mental healthcare costs are measured in euros and cover individual costs reimbursed through the Dutch basic insurance scheme. These include the costs of diagnosis-treatment combinations for mental healthcare, with or without a hospital stay, which cover mental health issues ranging from mild to severe. It is important to note that the variable only covers the period 2009 – 2022. In the period 2009 – 2013, mental healthcare costs are contained in a single measure by Statistics Netherlands. However, the variable definition changed in 2014 as mental healthcare costs were split into basic and specialized care. We add up the costs in basic and specialized care between 2014–2022 to compare them to the costs in 2009–2013. Panel a shows the average mental healthcare costs over time relative to diagnosis (at $t = 0$) for people with HIV (solid blue line) and matched controls (dashed gray line). Panel b makes an additional distinction between people with HIV and matched controls depending on stage at diagnosis: non-late (solid blue line and dashed light gray line, respectively) and late (solid orange line and dark gray dashed line, respectively).

An additional mechanism that is not discussed in the paper is the eligibility of DI. The paper is silent about the institutional setting of DI in the Netherlands. If DI is only granted to individuals with a diagnosis of a chronic disease, it is unsurprising that the fraction of recipients goes up after the HIV diagnosis is given. The paper would benefit from more details on the Dutch DI system. Furthermore, this raises the question, if the impact seen in this study also would apply to other countries with a less or more generous DI system for HIV patients.

We thank the reviewer for their comment and agree that the manuscript could benefit from a more detailed description of the disability insurance (DI) system and its eligibility requirements. We add the following paragraph to the beginning of section 2.5 on Disability insurance receipt:

New (lines 230 – 247): *Disability insurance (DI) in the Netherlands is administered by the Employee Insurance Agency (UWV), which determines the degree of disability of applicants based on their health status. During the first two years of illness, employers are legally obligated to continue paying individuals a minimum of 70% of their previous salary. Individuals who are unemployed or on temporary contracts can receive a Sickness Benefit in this period*

(Ziektewetuitkering). After two years of illness, the UWV assesses the individual's degree of disability based on their current and expected earning capacity. Individuals can enter one of two schemes: the WGA (Werkhervatting Gedeeltelijk Arbeidsgeschikten) for partially disabled individuals (35% - 80% degree of disability) or the IVA for fully disabled individuals (80% - 100% and permanent disability). Partially disabled individuals are expected to return to work, while receiving partial benefits, whereas fully disabled individuals are not. Self-employed individuals do not qualify for this scheme and instead must rely on private, voluntary insurance. The Dutch DI system is considered relatively generous by the OECD, although, in contrast with other European countries, it places a strong emphasis on reintegrating individuals with a disability into the labor force⁶. Generally, having an HIV diagnosis is not in itself a sufficient condition to qualify for disability insurance, with worsening mental health conditions being a more likely reason for sick leave and disability claims. However, severe cases of AIDS-defining conditions can lead to work disability, which might explain the results described below.

Therefore, it would be useful to know how much of the observed decline in employment, income and working hours is driven by individuals applying and receiving DI after being diagnosed with HIV. Table 2 indicates that the decline in employment of about 2.8 percentage points for HIV patients can entirely be explained by the increase of 3.6 percentage points higher rate of DI recipients. Similarly, part of the decline in labor income could be due to DI. It would be useful to make a decomposition of employment effect to see how much of the decline can be explained by more DI recipients. If it turns out that the decline in labor market outcome can be explained by an increase in DI recipients, it has important policy implications.

As described above, under the Dutch DI system, individuals are encouraged to return to work if they are not fully disabled, which implies that they might continue in employment but for fewer hours. To better highlight the relationship between disability insurance receipt and employment, we also make the following addition to section 2.7 Heterogeneity (now titled **2.7 Heterogeneity and mechanisms**) of the manuscript:

New (lines 417 – 429): *Another important mechanism that might explain the effects of an HIV diagnosis on labor market outcomes is the interaction between employment and disability*

insurance. In the Netherlands, individuals remain formally employed for the first two years of illness and continue being paid by their employers, after which their earnings capacity is assessed and they either enter a scheme for fully and permanently disabled individuals, or one for partly disabled individuals. Of the individuals with HIV in our sample who are receiving DI benefits two years after diagnosis, 78.8% have a degree of disability of at least 80%, 12.9% have a degree of disability between 45% and 80%, and 8.4% have a degree of disability of less than 45%. In addition, 36% of individuals who are receiving disability insurance or sickness benefits two years after diagnosis are also classified as employed for at least one month in the same year. Together, these figures suggest that, while disability insurance utilization may explain a part of the observed effect of an HIV diagnosis on income and employment, these effects are not entirely compensated by an increase in DI utilization.

Specific comments

In the introduction it should be highlighted under which assumption that the staggered Difference-in-Difference estimation will identify the causal effect. In section 4.4 these assumptions are discussed but it should also be stated in the introduction.

We agree with the reviewer and have made the following addition to the introduction:

New (lines 60 – 63): *The causal effect is identified under two main assumptions, namely the parallel trends assumption and the no anticipation assumption. The former states that the outcomes of individuals with HIV and those without would have evolved in parallel absent an HIV diagnosis. The latter states that individuals do not anticipate receiving an HIV diagnosis.*

In figure 3, the pre-trend for group diagnosed late is not flat. This could be an indication of anticipation effects.

Even though the pre-trends are not completely flat, the estimates for the pre-diagnosis effects of an HIV diagnosis on income in the late-diagnosed group are not statistically significant up to -5 years prior to diagnosis. We conduct an additional robustness check in which we evaluate the sensitivity of our estimates to violations of the parallel pre-trends assumptions (i.e., the honest difference-in-differences method proposed by Rambachan and Roth (2023)⁷).

We add the honest difference-in-difference analysis to the manuscript. Figure SI-2 is added to the Supplementary Information, and in section 2.6 Robustness checks we write:

New (lines 343 – 357): *To flexibly account for potential violations of the parallel trends assumption, we apply the honest difference-in-differences method proposed by Rambachan and Roth (2023)⁷. This method allows us to bound our estimated treatment effects under violations of parallel trends of different magnitudes, for all outcome variables. Panels a – d of figure SI-2 in the Supplementary Appendix show violations of parallel trends of various relative magnitudes and the corresponding bounds placed on the estimated treatment effects. We highlight in red the “breakdown” magnitude, which is the size of the violation at which we can no longer reject the null hypothesis that the treatment effect is equal to zero. As shown in the figures, these breakdown magnitudes are relatively large. For instance, the parallel trends violation for income can reach a magnitude of 1.5 times the maximum pre-diagnosis drift before our estimates become statistically insignificant. In other words, if the violation of parallel trends in the post-diagnosis period is no more than 1.5 times the maximum pre-diagnosis violation, our results remain statistically significant. For employment, work hours and disability insurance receipt, these magnitudes are similar, ranging from 1 to more than 1.5, which implies that our results are robust and that even a relatively large violation of parallel trends does not invalidate them.*

Figure SI-2: Honest difference-in-differences robustness check

This figure shows the results from running the Honest Difference-in-differences analysis following Rambachan and Roth (2022) on our main results. This allows us to bound our estimated treatment effects while flexibly accounting for potential violations of the parallel trends assumption. Thus, we can identify the "breakdown magnitude" of the violation, the value at which we can no longer reject the null hypothesis that our treatment effect is equal to zero. The X-axis in each figure shows different magnitudes of the violation (relative to the maximum pre-diagnosis drift). We highlight in red the breakdown magnitude.

It is also a bit surprising that the income effect is bigger for the non-late diagnosed group compared the late diagnosed group.

We acknowledge that this is an interesting and surprising result, which might be due to the somewhat larger likelihood that individuals diagnosed late have no income compared to individuals not diagnosed late (because of a lower probability of working). Indeed, 18.2% of individuals diagnosed late have no income one year before diagnosis, compared to 14.4% among those not diagnosed late. Therefore, the non-late diagnosed are more likely to experience a loss of income at the extensive margin, by losing their employment after an HIV diagnosis. However, it is important to note that the estimated treatment effects for income are not statistically different between the two groups, and, as such, the relative effect in Table 2 should be interpreted cautiously.

We also make the following addition to section 2.4 Income:

Old (lines 205– 208): *These estimates are not statistically significantly different from one another given that the 95% confidence intervals for late- and non-late diagnosed individuals overlap in Figure 3 and Table 2. Hence, people with HIV experience similar reductions in income regardless of the HIV disease stage at diagnosis.*

New (lines 215 – 221): *These estimates are not statistically significantly different from one another given that the 95% confidence intervals for late- and non-late diagnosed individuals overlap in Figure 3 and Table 2. Hence, people with HIV experience similar reductions in **absolute** income regardless of the HIV disease stage at diagnosis, **which might be explained by the fact that individuals with a timely diagnosis are more likely to be employed at baseline, implying larger potential losses: 18.2% of individuals diagnosed late have no income one year before diagnosis, compared to 14.4% of individuals not diagnosed late.***

The distinction between late and non-late diagnosis is interesting and the analyses show bigger effects for the individuals with late diagnosis. It is also clear that there are systematic differences between the two groups, where the late diagnosis group contains a higher share of individuals with unknown education and 1st generation immigrants. The paper does not explain why some individuals are diagnosed late in a universal health care system with relatively low health inequality.

Across the WHO European Region, the share of individuals diagnosed late has been increasing in recent years, with more than half of new diagnoses being made late⁸. This remains the case in the Netherlands as well, with 46% of HIV diagnoses in 2023 being late⁹. A Dutch study covering the period 1996 – 2014 suggests that factors associated with late-stage presentation included transmission through heterosexual intercourse, as well as region of origin^{9,10}. Similarly, a qualitative study conducted in the Amsterdam region has identified psychosocial factors (e.g., low perceived risk, stigma) and health system factors (e.g., perceived or actual limited accessibility to HIV testing facilities) as reasons for avoiding HIV testing, and thus as risk factors for a late diagnosis¹¹. This might especially be the case, for instance, for individuals with a migration background and unknown education (two groups which often overlap). Such individuals might face difficulties in adapting to a new healthcare system when arriving in the Netherlands due to barriers stemming from language, stigma or institutions.

We make the following addition to the manuscript, in section 3 Discussion:

Old (lines 325 – 330): *In this paper, we show that despite the universal accessibility of ART in the Netherlands, there is a significant negative impact of an HIV diagnosis on socioeconomic outcomes compared to a sample of matched controls. After an HIV diagnosis, individuals are 3.9 percent less likely to be employed, work 5.4 percent less FTE hours, have 8.9 percent lower annual incomes, and a 46.2 percent higher take-up of disability insurance (DI), compared to their pre-diagnosis outcomes. These effects vary by the disease stage at the time of diagnosis.*

New (lines 430 – 446): *In this paper, we show that despite the universal accessibility of ART in the Netherlands, there is a significant negative impact of an HIV diagnosis on socioeconomic outcomes compared to a sample of matched controls. After an HIV diagnosis, individuals are 3.9 percent less likely to be employed, work 5.4 percent less FTE hours, have 8.9 percent lower*

annual incomes, and a 46.2 percent higher take-up of disability insurance (DI), compared to their pre-diagnosis outcomes.

Even in a country with universal healthcare such as the Netherlands, health inequality remains an important issue. In the context of HIV, this can translate into a high share of individuals being diagnosed in a late or advanced stage. This can occur due to both psychosocial causes and (perceived or actual) barriers to healthcare services, with important factors including having a migration background, being a heterosexual male, and having a low income^{9,10}. A qualitative study conducted in the Amsterdam region has identified psychosocial factors, such as low perceived risk or perceived stigma, and health system factors, such as perceived limited accessibility to HIV testing facilities, as reasons for avoiding HIV testing, and thus as risk factors for a late diagnosis¹¹. Therefore, we also study whether the effects of receiving an HIV diagnosis vary by the disease stage at the time of diagnosis. For individuals who are diagnosed with late-stage HIV, we find a 6.3 percent decrease in employment, a 6.3 percent decline in FTE hours, 8.6 percent lower annual incomes, and a 60.5 percent higher utilization of DI.

References

1. Trickey, A. *et al.* Life expectancy after 2015 of adults with HIV on long-term antiretroviral therapy in Europe and North America: a collaborative analysis of cohort studies. *Lancet HIV* 10, e295–e307 (2023).
2. Tseng, A., Seet, J. & Phillips, E. J. The evolution of three decades of antiretroviral therapy: challenges, triumphs and the promise of the future. *Br J Clin Pharmacol* 79, 182–194 (2015).
3. Margolis, A. M., Heverling, H., Pham, P. A. & Stolbach, A. A Review of the Toxicity of HIV Medications. *Journal of Medical Toxicology* 10, 26–39 (2014).
4. Chawla, A. *et al.* A Review of Long-Term Toxicity of Antiretroviral Treatment Regimens and Implications for an Aging Population. *Infect Dis Ther* 7, 183–195 (2018).
5. *EACS European AIDS Clinical Society 1 EACS Guidelines 12.0.*
6. Hemmings, P. & Prinz, C. *Sickness and Disability Systems: Comparing Outcomes and Policies in Norway with Those in Sweden, the Netherlands and Switzerland.* <https://dx.doi.org/10.1787/c768699b-en> (2020) doi:10.1787/c768699b-en.
7. Rambachan, A. & Roth, J. A More Credible Approach to Parallel Trends. *Rev Econ Stud* 90, 2555–2591 (2023).
8. WHO Regional Office for Europe, E. C. for D. P. and Control. *HIV/AIDS Surveillance in Europe 2024 – 2023 Data.* <https://www.ecdc.europa.eu/en/publications-data/hiv-aids-surveillance-europe-2024-2023-data> (2024).
9. van Sighem, A. *et al.* *Monitoring Report 2024. Human Immunodeficiency Virus (HIV) Infection in the Netherlands.* (Stichting hiv monitoring, Amsterdam, 2024).
10. Op De Coul, E. L. M. *et al.* Factors associated with presenting late or with advanced HIV disease in the Netherlands, 1996–2014: results from a national observational cohort. *BMJ Open* 6, e009688 (2016).
11. Bedert, M. *et al.* Understanding Reasons for HIV Late Diagnosis: A Qualitative Study Among HIV-Positive Individuals in Amsterdam, The Netherlands. *AIDS Behav* 25, 2898–2906 (2021).

Point by Point Responses to Comments from Referee 2

We sincerely thank the referee for their thorough and constructive comments on our paper, “Labor Market Outcomes of People with HIV Pre- and Post-Diagnosis”. We greatly appreciate the time and effort spent on providing feedback, which has been instrumental in improving the quality of our paper.

Below, we provide a point-by-point response to all your comments. In our responses, bold text represents your original comments, plain text provides our answers, and italicized text includes direct quotes from the revised manuscript. Note that we marked changes to the text in the manuscript in red.

Comment:

There are differences observed with regards to work/income even prior to HIV diagnosis, which, as the authors also note, could be due to unmatched differences with regards to preferred types of employment or other social impacts for people in gender minorities. Given that the majority of HIV infections in high resourced settings occur among men who have sex with men, an additional sub-analysis for men might include a control group of men who have sex with men without HIV; or perhaps a group of men with another STI (perhaps a rectal STI). If such data sources are available, the authors might want to consider this comparison, as it would match more closely the sample of people with HIV (for men), strengthening the claim of causality and assessment of size of impact. Sources could be surveys or medical records, even though the sample size may not be as large.

We thank the reviewer for their comment. We acknowledge that our study would benefit from being able to match, for instance, men who have sex with men diagnosed with HIV with MSM from the general Dutch population. However, because of privacy regulations, these data are not available in the administrative records of Statistics Netherlands. Similarly, while it would be helpful to match on the presence of rectal STIs, these detailed data are also not available in Statistics Netherlands.

The issue of matching on sexual behaviour or orientation is also raised by Reviewer 3, so we would like to address both reviewers’ comments by making the following addition to section 3 Discussion:

Old (lines 379 – 386): *Our study has two important limitations. First, we are unable to control for unobserved time-varying characteristics of people with HIV and their matched controls. This might explain why we see a small deviation in the pre-diagnosis trends in income between these two groups, raising some concerns about the comparability of our matched control group. Despite this, we provide evidence that the parallel trends assumption holds in Figures 1–4. Second, the generalizability of our results might be limited to other high-income countries. The Netherlands provides a “best-case” scenario in which to measure the effects of HIV, as HIV care is widely available. Therefore, it is difficult to extrapolate these results to low- and middle-income settings.*

New (lines 497 – 507): *Our study has two important limitations. First, we are unable to control for unobserved time-varying characteristics of people with HIV and their matched controls. This might explain why we see a small deviation in the pre-diagnosis trends in income between these two groups, raising some concerns about the comparability of our matched control group, for example on sexual behavior and/or orientation. Due to data limitations, we cannot match individuals on these characteristics, even though they could influence employment and lifestyle choices, which, in turn, can impact labor market outcomes. Despite this, we provide evidence that the parallel trends assumption holds in Figures 1–4. Second, the generalizability of our results might be limited to other high-income countries. The Netherlands provides a “best-case” scenario in which to measure the effects of HIV, as HIV care is widely available. Therefore, it is difficult to extrapolate these results to low- and middle-income settings.*

Point by Point Responses to Comments from Referee 3

We sincerely thank the referee for their thorough and constructive comments on our paper, “Labor Market Outcomes of People with HIV Pre- and Post-Diagnosis”. We greatly appreciate the time and effort spent on providing feedback, which has been instrumental in improving the quality of our paper.

Below, we provide a point-by-point response to all your comments. In our responses, bold text represents your original comments, plain text provides our answers, and italicized text includes direct quotes from the revised manuscript. Note that we marked changes to the text in the manuscript in red.

In the abstract I would state the restriction to the period 2010-2022 – I think this strengthens the analysis because the early period of the HIV epidemic was very different and inferences about employment based on that period will have much less relevance now.

We have made the following edit to the abstract:

Old (lines 11 – 13): *A causal effect is estimated by comparing outcomes of people with HIV to a matched control group in a difference-in-difference design*

New (lines 12 – 13): *A causal effect is estimated by comparing outcomes of people with HIV **diagnosed between 2010 and 2022** to a matched control group in a difference-in-difference design.*

In the introduction, the background of the question, the aims and broad methodology are well explained, but there is no statement of rationale for the work saying why it is important to address this question.

We thank the reviewer for their comment. We have made the following edit to section 1 Introduction:

Old (lines 55 – 60): *The Netherlands is a high-income country with universal healthcare access including ART¹⁶ and a strong social support system and relatively low health inequalities. Therefore, the Netherlands likely represents a “best-case” scenario in which changes in labor*

market outcomes after an HIV diagnosis cannot be explained by insufficient access to relevant healthcare and social services. The impact of receiving an HIV diagnosis on labor market outcomes may be larger in settings with less social support and poorer access to care.

New (lines 64 – 72): *By identifying the causal effects of receiving an HIV diagnosis on labor market outcomes, we highlight the need for continued efforts of prevention and early detection of HIV, as well as for targeted interventions aimed at helping people with HIV reintegrate in the labor market. The Netherlands is a high-income country with universal healthcare access including ART¹⁶, a strong social support system and relatively low health inequalities. Therefore, the Netherlands likely represents a “best-case” scenario in which changes in labor market outcomes after an HIV diagnosis cannot be explained by insufficient access to relevant healthcare and social services. The impact of receiving an HIV diagnosis on labor market outcomes may be larger in settings with less social support and poorer access to care.*

Is there a general methods paper for the ATHENA cohort that could be referenced for further details of cohort methodology and regulatory processes?

The profile of the ATHENA cohort is described by Boender et al. (2018)¹. We added this reference to the manuscript, see below:

Old (lines 401 – 403): *Data collection was initiated in 1998 and data are prospectively collected in the ATHENA (AIDS Therapy Evaluation in the Netherlands) cohort, which represents over 98 percent of all people with HIV in care in the Netherlands.*

New (lines 522 – 524): *Data collection was initiated in 1998 and data are prospectively collected in the ATHENA (AIDS Therapy Evaluation in the Netherlands) cohort, which represents over 98 percent of all people with HIV in care in the Netherlands¹.*

Some points regarding the results. Table 1 – why is there no missing data for migration status as there is for education? Table 2 – please explain the definition of the effect size in the footnote.

First, migration status is available in the registry data from Statistics Netherlands for all people registered in the Netherlands, based on the individual's birthplace or their parents' birthplace. By contrast, education data only started being systematically collected from 1999, while education levels obtained earlier are imputed from surveys. This implies that individuals from older cohorts might have missing education data. Similarly, the educational attainment of individuals with degrees obtained abroad is not collected systematically.

Second, we have made the following edit to the footnote of Table 2 to explain the definition of the effects size:

Old (lines 275 – 276): *The effect size is defined as the size of the estimated treatment effect relative to the pre-diagnosis mean.*

New (lines 322 – 320): *For each outcome, the effect size is defined as the magnitude of the point estimate of the treatment effect divided by the pre-diagnosis mean of that outcome for people with HIV and multiplied by one hundred.*

Relating to Figure 1 – please explain why a general downward trend is observed in employment with time – would this be expected with age, or does it reflect methodological aspects of the comparison?

Indeed, we expect that the downward trend in employment across all groups is due to ageing, as individuals reach (early) retirement age or experience work-limiting disabilities that cause them to leave the labor market. Considering that our youngest cohorts are at least 21 years old seven years prior to diagnosis, we expect that there is relatively little inflow into employment. Taken together, these facts point to a net outflow from employment into unemployment, which helps to explain the downward trend observed.

We acknowledge that this is an interesting point, and made the following addition to section 2.2 Employment:

Old (lines 96 – 99): *The outcomes of matched controls are captured at the same age as their matched person with HIV. Trends in employment run in parallel between the two groups in the pre-diagnosis period, but after diagnosis a large decline in employment is observed for people with HIV relative to matched controls.*

New (lines 107 – 113): *The outcomes of matched controls are captured at the same age as their matched person with HIV. Both groups experience a downward trend in employment over time, which is likely due to individuals reaching (early) retirement age, or the higher probability of experiencing work-limiting disabilities or exiting the labor market at older ages. These trends run in parallel between the two groups in the pre-diagnosis period, but after diagnosis a large decline in employment is observed for people with HIV relative to matched controls.*

Also of interest to comment on (in results/discussion) is the fact that when comparing people with and without HIV, the widening of the pre-diagnosis gap occurs from diagnosis to two years afterwards – following this the lines run parallel again, suggesting quite an immediate effect of HIV on employment after diagnosis, rather than a continually widening gap subsequently. In fact, graph (d) implies some ‘catch-up’ in the late diagnosis subgroup over time. Similar patterns are apparent for most of the other outcomes in the remaining figures – please give your interpretation of these patterns.

We identify two potential reasons for this pattern. First, it could be the result of selective attrition, with unhealthier individuals, whose labor market outcomes would be expected to be worse, dropping out of the sample over time (for example, due to death). Second, the pattern could be caused by an amelioration in health, as we would expect the health status of individuals with a late-stage diagnosis to improve once they initiate ART and achieve viral suppression. To test for the first explanation, we conduct an additional robustness check in which we re-run our analysis but restrict the sample to individuals who are still alive by the end of the panel, in the year 2022. This is introduced in table SI-1 under column (5), which shows that the results are essentially unaffected. We also highlight this change below.

We make the following edit to section 2.2 Employment:

Old (lines 127 – 131): *People with HIV experience a decline in employment starting within one year after diagnosis regardless of the stage of the disease at diagnosis. However, the effects are more pronounced for people with HIV who are diagnosed late, who experience a persistent and larger decline in employment starting in the year of diagnosis.*

New (lines 141 – 147): *People with HIV experience a decline in employment starting within one year after diagnosis regardless of the stage of the disease at diagnosis. The effects are more pronounced for people with HIV who are diagnosed late, who experience a persistent and larger decline in employment starting in the year of diagnosis. However, late diagnosed individuals also experience a narrowing of the gap over time, which might be the result of selective attrition (unhealthier individuals leaving the sample due to death) and/or of an improvement in health following the initiation of antiretroviral therapy.*

Additionally, we introduce the following edit to section 2.6 Robustness checks:

Old (lines 264 – 267): *Therefore, as a third test, we restrict the sample to the cohorts 2010–2015 (3,720 people with HIV compared to 5,960 in the baseline analysis) that can be observed across all 7 years pre- and post-diagnosis. Column (4) shows that estimates are similar to the main results when restricting the analyses to these cohorts.*

New (lines 308 – 315): *Therefore, as a third test, we restrict the sample to the cohorts 2010–2015 (3,720 people with HIV compared to 5,960 in the baseline analysis) that can be observed across all 7 years pre- and post-diagnosis. Column (4) shows that estimates are similar to the main results when restricting the analyses to these cohorts. Finally, we conduct a fourth test, which consists of excluding from the sample individuals who pass away before the final year of the panel, 2022. This removes 319 people with HIV and 1,389 matched controls. Column (5) of Table SI-1 shows that that the estimated treatment effects are almost unchanged.*

Table SI-1: Robustness check results

	Main results	Replacing with 0/1	Replacing with last observed	2010-2015	Only individuals alive by 2022	Not-yet diagnosed
	(1)	(2)	(3)	(4)	(5)	(6)
Panel A: Employment						
ATT	-0.028*** (0.005)	-0.041*** (0.005)	-0.032*** (0.005)	-0.024*** (0.006)	-0.028*** (0.005)	-0.054*** (0.010)
Pre-diagnosis mean	0.719	0.719	0.719	0.722	0.727	0.719
Observations	65,560	65,560	65,560	40,920	63,852	5,960
Panel B: Work hours						
ATT	-0.035*** (0.005)	-0.047*** (0.005)	-0.039*** (0.005)	-0.031*** (0.005)	-0.035*** (0.005)	-0.052*** (0.009)
Pre-diagnosis mean	0.650	0.650	0.650	0.662	0.657	0.650
Observations	61,974	61,974	61,974	38,896	60,361	5,634
Panel C: Income						
ATT	-3,583.8*** (543.2)	-4,313.1*** (544.1)	-3,863.4*** (543.5)	-3,531.7*** (652.5)	-3468.5*** (567.3)	-3,671.5*** (757.8)
Pre-diagnosis mean	40,135.2	40,135.2	40,135.2	41,924.3	40,482.2	40,135.2
Observations	64,416	64,416	64,416	40,326	62,734	5,856
Panel D: Disability Insurance						
ATT	0.036*** (0.004)	0.056*** (0.004)	0.037*** (0.004)	0.036*** (0.004)	0.035*** (0.004)	0.039*** (0.006)
Pre-diagnosis mean	0.078	0.078	0.078	0.077	0.075	0.078
Observations	65,560	65,560	65,560	40,920	63,852	5,960

Notes: This table shows the average treatment effects on the treated (ATT) for our robustness checks, estimated using the Callaway-Sant'Anna estimator and aggregation procedure for staggered difference-in-differences [1]. Standard errors are reported in parentheses. The results are reported for each of our main outcomes (employment, work hours, income and disability insurance receipt) and only for the overall population. Column (1) shows our main results, as seen in column (1) of Table 1. Column (2) shows the results from running the analysis after replacing missing post-diagnosis outcomes with 0. Column (3) shows the results after replacing missing outcomes with the last observed value. Column (4) shows the results after restricting the sample to individuals diagnosed between 2010 and 2015. Column (5) shows the results after restricting the sample to individuals who are still alive by the year 2022 (the end of the panel). Column (6) reports the results from using the not-yet diagnosed control group, rather than the general population (the never-diagnosed). The pre-diagnosis mean shows mean employment, work hours, income and disability insurance receipt for the group of people with HIV, measured in the year before diagnosis. The number of observations is defined as the number of individuals observed in the year prior to diagnosis. Robust and asymptotic standard errors are reported in parentheses. $p < 0.10$, ** $p < 0.05$, *** $p < 0.01$.

In addition, the contracted scales on the graphs perhaps overstate some differences a little –e.g. a few percentage points for employment on Figure 1, suggesting a relatively modest difference. However, differences do appear greater for some of the other employment measures. From this perspective, it would be useful to have some more information on how the magnitude of difference for HIV compares to that for other chronic conditions where analyses, outcomes and time period are broadly comparable – this is touched on in the discussion, it would be nice to see this expanded somewhat.

We acknowledge that it is interesting to expand the comparison with other findings from the literature. We make the following edits to section 3 Discussion:

Old (lines 336 – 338): *The negative effects for the late-diagnosed group in this paper are comparable to income and earnings losses that are observed after diagnosis with types of cancer with high survival rates (e.g., thyroid cancer, prostate cancer or breast cancer).*

New (lines 452 – 458): *The negative effects for the late-diagnosed group in this paper are comparable to income and earnings losses that are observed after diagnosis with types of cancer with high survival rates (e.g., thyroid cancer, prostate cancer or breast cancer), such as a 9–10% reduction in income and a 2–4% reduction in the employment probability in the years after diagnosis¹⁹. Similarly, our results point in a similar direction to those identified in previous studies of the employment and work hour effects of other chronic conditions, such as hypertension, high blood cholesterol and diabetes in Australia and several European countries^{2,3}.*

Although the investigators matched on a number of demographic factors (sex, age, migration status, education), they did not match on sexuality. I assume such data are not available, but this could be relevant, as people with HIV are more likely to be gay/bisexual men/MSM compared to the general population. This subgroup could have different employment profiles (leading to an under or over-estimate of HIV versus non-HIV differences). For example they are perhaps less likely to have caring responsibilities, which impact employment options and choices. They may also have elevated levels of some specific lifestyle factors such as smoking and drug use which may also act as confounders in

the comparison. Although additional confounding is mentioned in the discussion, there is no consideration of specific factors, which I think is needed.

We thank the reviewer for their comment, which was also brought up by Reviewer 2. We acknowledge that our study would benefit from being able to match, for instance, men who have sex with men diagnosed with HIV with MSM from the general Dutch population. However, because of privacy regulations, these data are not available in the administrative records of Statistics Netherlands. We rewrote section 3 Discussion to acknowledge these data issues:

Old (lines 379 – 386): *Our study has two important limitations. First, we are unable to control for unobserved time-varying characteristics of people with HIV and their matched controls. This might explain why we see a small deviation in the pre-diagnosis trends in income between these two groups, raising some concerns about the comparability of our matched control group. Despite this, we provide evidence that the parallel trends assumption holds in Figures 1–4. Second, the generalizability of our results might be limited to other high-income countries. The Netherlands provides a “best-case” scenario in which to measure the effects of HIV, as HIV care is widely available. Therefore, it is difficult to extrapolate these results to low- and middle-income settings.*

New (lines 498 – 508): *Our study has two important limitations. First, we are unable to control for unobserved time-varying characteristics of people with HIV and their matched controls. This might explain why we see a small deviation in the pre-diagnosis trends in income between these two groups, raising some concerns about the comparability of our matched control group, for example on sexual behavior and/or orientation. Due to data limitations, we cannot match individuals on these characteristics, even though they could influence employment and lifestyle choices, which, in turn, can impact labor market outcomes. Despite this, we provide evidence that the parallel trends assumption holds in Figures 1–4. Second, the generalizability of our results might be limited to other high-income countries. The Netherlands provides a “best-case” scenario in which to measure the effects of HIV, as HIV care is widely available. Therefore, it is difficult to extrapolate these results to low- and middle-income settings.*

The section on heterogeneity is useful - I would also like to suggest two further stratifications. First, based on period of diagnosis, because even in this restricted 12-year period (2010-2022) there have been continual improvements in treatment as well as potential changes in management of the psychosocial aspects of HIV. Looking at those with earlier and later diagnoses within this period would shed light on whether employment outcomes relative to those without HIV are improving for those diagnosed more recently.

We thank the reviewer for their suggestion. Indeed, we recognize the importance of studying the heterogeneity of our results by recency of diagnosis. To this end, we conduct an additional analysis, which we include in Supplementary Information file as figures SI-15–SI-18. These figures follow the same format as figures SI-3–SI-14 but instead stratify the sample by whether individuals were diagnosed in the period 2010–2015 or the period 2015–2022. They are shown below.

We also replace the word “finally” with the word “next” in line 313 in section 2.7, and we also add the following paragraph to the section:

New (lines 387 – 395): *We also conduct a heterogeneity analysis by recency of diagnosis, based on two groups, namely individuals diagnosed between 2010 and 2015, and individuals diagnosed between 2015 and 2022. This distinction highlights continual improvements in treatment and in the management of psychosocial aspects of HIV. We note that despite the observed differences in trends between the two periods observed across all four outcomes, the general patterns described in the main results remain. In all outcomes, there is a large drop in employment, work hours and income, and an increase in disability insurance utilization, after HIV diagnosis, among individuals diagnosed both before and after 2015. This concordance likely reflects similar approaches in the management of HIV between these periods.*

Figure SI-15: Employment by period

Note: Employment is defined as having income from employment for at least one month in a year. Panel (a) shows the proportion of individuals in employment for at least 1 month over time relative to diagnosis (at $t = 0$), for people diagnosed before the year 2015 and their matched controls. Panel (b) shows the same outcome for individuals diagnosed in the year 2015 or later. Matched controls are assigned to the same year of diagnosis as their respective matched person with HIV.

Figure SI-16: Work hours by period

Note: Hours worked is defined in full-time equivalents (FTE), which implies that a value of one represents a full-time job, and zero that the individual does not work. Panel (a) shows the work hours relative to FTE over time relative to diagnosis (at $t = 0$), for people diagnosed before the year 2015 and their matched controls. Panel (b) shows the same outcome for individuals diagnosed in the year 2015 or later. Matched controls are assigned to the same year of diagnosis as their respective matched person with HIV.

Figure SI-17: Income by period

Note: Income is defined as gross annual income from work and self-employment. Panel (a) shows the yearly income from work (in 2015 euros) over time relative to diagnosis (at $t = 0$), for people diagnosed before the year 2015 and their matched controls. Panel (b) shows the same outcome for individuals diagnosed in the year 2015 or later. Matched controls are assigned to the same year of diagnosis as their respective matched person with HIV.

Figure SI-18: Disability insurance receipt by period

Note: Disability insurance (DI) take-up is defined as receiving disability or sickness benefits for at least one month in a given year. Panel (a) shows the proportion of individuals receiving DI or sickness benefits for at least 1 month over time relative to diagnosis (at $t = 0$), for people diagnosed before the year 2015 and their matched controls. Panel (b) shows the same outcome for individuals diagnosed in the year 2015 or later. Matched controls are assigned to the same year of diagnosis as their respective matched person with HIV.

Second, related to my earlier comment - based on sex, because childcare responsibilities are likely to impact considerably on employment, and certain subgroups affected by HIV (MSM) may be less likely to have these responsibilities (see my previous comment) it would be useful to look at the analysis in the female subgroup particularly. Still on this theme, another issue is that having HIV may impact reproductive choices, which would impact employment patterns.

We thank the reviewer for their helpful suggestion. As above, we recognize the importance of sex in understanding the effects of an HIV diagnosis. As such, we conduct an additional analysis, also included in the Supplementary Information file, as figures SI-19 – SI-22. These figures, following the same format as above, disaggregate the sample by registered gender, and are shown below.

We introduce the following paragraph to section 2.7:

New (lines 396 – 402): *Finally, we stratify the sample by registered gender, which could reflect differences in, for instance, childcare responsibilities, reproductive choices, as well as underlying differences in labor market outcomes. Across all outcomes, a clear pattern emerges: women experience worse labor market outcomes after an HIV diagnosis than men, perhaps reflecting differences in caring responsibilities or different levels of labor market attachment. These discrepancies are also likely explained by a difference in the prevalence of late-stage diagnoses in the female group, considering that the share of women with a late-stage diagnosis is 51.3%, compared to 41.3% among men.*

Figure SI-19: Employment by gender

Note: Employment is defined as having income from employment for at least one month in a year. Panel (a) shows the proportion of individuals in employment for at least 1 month over time relative to diagnosis (at $t = 0$), for individuals registered as males by Statistics Netherlands. Panel (b) shows the same outcome for individuals registered as females. Matched controls are assigned to the same year of diagnosis as their respective matched person with HIV.

Figure SI-20: Work hours by gender

Note: Hours worked is defined in full-time equivalents (FTE), which implies that a value of one represents a full-time job, and zero that the individual does not work. Panel (a) shows the work hours relative to FTE over time relative to diagnosis (at $t = 0$), for individuals registered as males by Statistics Netherlands. Panel (b) shows the same outcome for individuals registered as females. Matched controls are assigned to the same year of diagnosis as their respective matched person with HIV.

Figure SI-21: Income by gender

Note: Income is defined as gross annual income from work and self-employment. Panel (a) shows the yearly income from work (in 2015 euros) over time relative to diagnosis (at $t = 0$), for individuals registered as males by Statistics Netherlands. Panel (b) shows the same outcome for individuals registered as females. Matched controls are assigned to the same year of diagnosis as their respective matched person with HIV.

Figure SI-22: Disability insurance receipt by gender

Note: Disability insurance (DI) take-up is defined as receiving disability or sickness benefits for at least one month in a given year. Panel (a) shows the proportion of individuals receiving DI or sickness benefits for at least 1 month over time relative to diagnosis (at $t = 0$), for individuals registered as males by Statistics Netherlands. Panel (b) shows the same outcome for individuals registered as females. Matched controls are assigned to the same year of diagnosis as their respective matched person with HIV.

In the methods the selection processes for inclusion are outlined. 15-20% of the people diagnosed between 2010 and 2022 are excluded because employment and disability insurance receipt data are not available for the entire period of seven years before diagnosis. Is this group likely to be different in terms of employment or other factors than those with data? These potential biases should be considered in the discussion.

We thank the reviewer for their comment. We acknowledge the possibility that the individuals who are excluded due to their employment and disability insurance data not being available for seven years prior to diagnosis may be different from the final analysis sample. Indeed, they are, on average, younger at diagnosis than the individuals who remain in the sample (37.8 years versus 43.6 years), less likely to be male (73.7% versus 87.6%) and more likely to have a first generation migration background (86.6% versus 23.9%). While we cannot rule out the possibility that including these individuals in the analysis would change our results, this selection is made to have a balanced panel in the pre-diagnosis period.

We also add the following to section 4 Methods:

Old (lines 429 – 430): *The sample is further restricted to 5,960 individuals whose employment and disability insurance receipt data is available for the entire period of seven years before diagnosis.*

New (lines 550 – 555): *The sample is further restricted to 5,960 individuals whose employment and disability insurance receipt data is available for the entire period of seven years before diagnosis. Individuals who are removed from the sample in this way are younger than those who remain (37.8 years versus 43.6 years), less likely to be male (73.7% versus 87.6%) and more likely to have a first generation migration background (86.6% versus 23.9%). This selection step is done in order to maintain a balanced panel in the pre-diagnosis period.*

References

1. Boender, T. S. *et al.* AIDS Therapy Evaluation in the Netherlands (ATHENA) national observational HIV cohort: cohort profile. *BMJ Open* **8**, e022516 (2018).
2. Zhang, X., Zhao, X. & Harris, A. Chronic diseases and labour force participation in Australia. *J Health Econ* **28**, 91–108 (2009).
3. Polanco, B., Oña, A., Sabariego, C. & Pacheco Barzallo, D. Chronic health conditions and their impact on the labor market. A cross-country comparison in Europe. *SSM Popul Health* **26**, 101666 (2024).

Point by Point Responses to Comments from Referee 1

We sincerely thank the referee for their thorough and constructive comments on our revised manuscript, “Labor Market Outcomes of People with HIV Pre- and Post-Diagnosis”. We greatly appreciate the time and effort spent on reviewing the manuscript, as well as providing additional feedback, which has been instrumental in improving the quality of our paper.

Below, we provide a point-by-point response to all your comments. In our responses, bold text represents your original comments, plain text provides our answers, and italicized text includes direct quotes from the revised manuscript. Note that we marked changes to the text in the manuscript in red.

1. Mental Health Expenditures and Parallel Trends Assumption: I appreciate the inclusion of analyses on mental health expenditures in the current version. However, as noted by the authors in line 408, the trends in mental health care expenditures do not follow parallel trends. This is concerning, as it violates the assumption of parallel trends, which is fundamental to causal inference in event study designs. Consequently, the authors have chosen to present only descriptive evidence for mental health expenditures, which is a reasonable decision.

Nonetheless, this raises questions about the validity of the analyses for other outcome variables—namely employment, hours worked, income, and disability insurance (DI). If the HIV and control groups exhibit non-parallel pre-trends in mental health expenditures, it suggests that the groups may differ systematically even before diagnosis, with the HIV group potentially experiencing deteriorating mental health prior to diagnosis.

To address this concern, the authors should provide evidence that the estimated effects on labor market outcomes are not driven by these differential trends in mental health. One possible approach would be to construct a matched control group that exhibits similar trends in mental health expenditures. For example, matching on the level of mental health expenditures in periods -7 and -1 could help ensure comparability.

We thank the reviewer for their remark and acknowledge that non-parallel trends in mental healthcare costs before diagnosis may raise concerns about the validity of the parallel trends assumption for labor market outcomes. If individuals experience more mental health issues in the period leading up to an HIV diagnosis, these may affect their labor market outcomes as well in the pre-diagnosis period.

In a previous analysis, we considered mental healthcare costs from seven years prior to diagnosis to seven years after for the entire sample. We have taken a few steps to examine the extent of non-parallel trends in mental healthcare consumption as seen in that analysis. First, in Figure SI-23, panels a and b (below), we show the yearly proportion of individuals that incurred any mental health care costs in the four years prior and seven years post diagnosis for people with HIV diagnosed starting in 2013 and matched controls (i.e., the extensive margin). We focus on this period to ensure a balanced sample before and after diagnosis, since data on mental health care costs were only available from 2009 onwards. The figure clearly shows parallel trends before diagnosis, suggesting that the non-parallel trends in mental healthcare expenditures stem from existing patients increasing their use of mental health care services (i.e., the intensive margin), rather than new patients.

Second, given that the non-parallel trends in mental health care costs seem to be driven by a stable number of patients, we explored the hypothesis that this is driven by outliers in mental health care consumption. Specifically, we examined the sensitivity of the parallel trend plots for mental health care consumption on the intensive margin by completely removing from the sample outliers, i.e., individuals with mental health care expenditures above the 97.5 percentile of consumers at any point in the pre-diagnosis period (183 dropped individuals, 0.4% of the individuals with non-missing costs at t-1). We focus on outliers in the pre-diagnosis period to avoid excluding individuals whose mental healthcare costs increased due to an HIV diagnosis. Figure SI-23, panels c and d, shows that pre-trends are relatively parallel, and that individuals with an HIV diagnosis experienced a much larger increase in mental health care expenditures after diagnosis than the matched controls. This evidence is in line with a positive impact of an HIV diagnosis on mental healthcare consumption at the intensive margin. We note that the rise in mental health care costs at t=0 for the control group can be explained by the exclusion of outliers in the pre-diagnosis period, which mechanically increases costs in the post-period also for the control group.

These two additional analyses showed that we observe parallel trends for mental health care expenditures at the extensive margin, and that trends are more parallel at the intensive margin when excluding outliers in the pre-diagnosis period. One may still be concerned about the impact of removing outliers for our results on labor market outcomes. However, since we only removed 0.4% percent of individuals from the main sample, the effect on reported results for labor market outcomes is likely to be minimal. To support our conclusion, we also included figures for employment, work hours and disability insurance utilization after removing the outliers from the sample below, in Figure R1 (included below but not in the manuscript). As expected, these figures

show parallel trends similar to our main results. We also performed a bounding exercise, where instead of dropping the outliers from the sample, we replaced their main outcomes with a worst-case value (0 in the case of employment, work hours and income and 1 for DI), in Figure R2 (included below by not in the manuscript). These figures show that results and parallel trends are similar even if the removed individuals experienced worst-case outcomes.

Apart from this additional evidence on parallel trends in mental health and labor market outcomes prior to diagnosis, please note that the results in Appendix figures SI-1 show that our main results on labor market outcomes are robust to relatively large violations of parallel trends using the honest difference-in-differences approach. Furthermore, we also report results that only use the not-yet HIV diagnosed group as the control group in column 6 of table SI-1. This exercise compares people with HIV to a more similar control group (people who are diagnosed with HIV at a later date), which minimizes the risk of a parallel trends violation by reducing differences both in the levels of care consumption and in unobserved, time-varying characteristics and behaviors between the treated and control groups. These estimates are consistent with our main sample estimates. The new results combined with these robustness tests assert the credibility of the parallel trends assumption for our main outcome variables.

We have considered the referee's suggestion for making the control group more comparable to people with HIV by matching on mental healthcare expenditures in the pre-diagnosis period. However, we decided against this approach. Mental healthcare costs may increase before diagnosis if individuals seek mental health care in anticipation of a diagnosis (for example, because of worries about HIV acquisition). In this spirit, matching on mental health care expenditures before diagnosis would suffer from the "bad control" problem, that is, because an increase in mental healthcare consumption is an early result of the forthcoming HIV diagnosis, it would be similar to matching on an outcome. We would ideally like to match on plausibly exogenous characteristics that have impact on the chance of acquiring HIV, like sexual identity, sexual preference and behavioural characteristics such as substance use, the use or non-use of biomedical HIV prevention tools and/or the number of sexual partners. As mentioned in lines 506 – 512 of the manuscript, these data are however unavailable.

We replace Figure SI-23 in the previous version of the manuscript with the following figure:

Figure SI-23: Mental healthcare costs - extensive and intensive margins

Note: The figure shows mental healthcare costs, as reimbursed through the Dutch basic insurance scheme. These include the costs of diagnosis-treatment combinations for mental healthcare, with or without a hospital stay, which cover mental health issues ranging from mild to severe. It is important to note that the variable only covers the period 2009 – 2022. For this reason, we focus on cohorts diagnosed in the period 2013 - 2022, such that they can be observed for at least four years before diagnosis, and report values from four years before diagnosis to 7 years after diagnosis. In the period 2009 – 2013, mental healthcare costs are contained in a single measure by Statistics Netherlands. However, the variable definition changed in 2014 as mental healthcare costs were split into basic and specialized care. We add up the costs in basic and specialized care between 2014–2022 to compare them to the costs in 2009–2013. Panel a shows the share of individuals with any positive amount of mental health care expenditures (i.e., the extensive margin) over time relative to diagnosis (at $t = 0$) for people with HIV (solid blue line) and matched controls (dashed gray line). Panel b makes an additional distinction between people with HIV and matched controls depending on the stage of HIV at diagnosis: non-late (solid blue line and dashed light gray line, respectively) and late (solid orange line and dark gray dashed line, respectively). Panels c and d show mean expenditures in euros per year. This is shown after removing from the sample individuals whose care expenditures are above the 97.5th percentile of consumers (excluding individuals who do not consume any mental healthcare) at any point in the four years prior to diagnosis. This removes from the sample 30 people with HIV (0.8% of the 3901 people with HIV with non-missing mental healthcare costs at $t-1$) and 153 controls (0.4% of the 38,801 with non-missing expenditures at $t-1$). Panel c shows this for all people with HIV and matched controls. Panel d separates the sample by HIV stage at diagnosis, as described above. The small increase observed at the time of diagnosis for the matched controls is caused by the fact only outliers in the pre-diagnosis period are removed from the sample.

We make the following addition to Section 2.7 of the manuscript to reflect these results:

Old (lines 403 – 416): *To study potential mechanisms through which an HIV diagnosis might impact labor market outcomes, we examine individuals' mental healthcare expenditures, as*

identified in the data by basic health insurance reimbursements, before and after HIV diagnosis. All individuals in the Netherlands are required by law to take up basic health insurance, which covers mental healthcare, so we expect that underreporting is not a concern. We present this data as a descriptive analysis because of concerns about violations of the parallel trends assumption. Figure SI-23 shows mental healthcare expenditures as reimbursed through the basic health insurance system. We find that people with HIV generally have higher mental healthcare expenditures than the matched controls, both in general (Figure SI-23, panel a) and when separating the sample by stage of HIV at diagnosis (figure SI-23, panel b). Interestingly, individuals diagnosed late follow a different trend compared to those not diagnosed late in the pre-diagnosis period. However, whereas those with a timely diagnosis experience a persistent increase in mental healthcare costs which seems to accelerate post-diagnosis, individuals diagnosed late experience a sharp increase in costs around the time of diagnosis, followed by a large decline.

New (lines 403 – 422): To study potential mechanisms through which an HIV diagnosis might impact labor market outcomes, we examine individuals' mental healthcare expenditures, as identified in the data by basic health insurance reimbursements, before and after HIV diagnosis. All individuals in the Netherlands are required by law to take up basic health insurance, which covers mental healthcare, so we expect that underreporting is not a concern. Figure SI-23 shows mental healthcare expenditures as reimbursed through the basic health insurance system before and after an HIV diagnosis. Panels a and b show the share of individuals with any mental healthcare costs (the extensive margin). These figures display strong parallel trends in the pre-diagnosis period, indicating that the share of people with HIV utilizing mental healthcare evolves at the same rate as the matched controls. People with HIV experience an increase in the probability of consuming any mental health care after HIV diagnosis, whereas this is not observed for the matched controls. In panels c and d, we examine annual expenditures for mental health care (the intensive margin). These figures also show relatively flat parallel trends. Note that outliers, i.e., individuals with mental healthcare expenditures above the 97.5th percentile in any year in the pre-diagnosis period, are left out (183 dropped individuals, 0.4% of the sample with non-missing healthcare costs at t-1) because they have a large effect on consumption trajectories. While both groups experience an increase at the time of diagnosis due to only removing pre-diagnosis outliers, the discontinuity is much more pronounced for the people with HIV. These results provide evidence of a sharp rise in mental healthcare utilization around the time of diagnosis, suggesting that a deterioration in mental health is a potential mechanism for the effects of HIV on labor market outcomes.

Figure R-1: Main outcomes - dropping outliers

Note: This figure shows mean outcomes after dropping from the sample individuals defined as mental healthcare expenditure outliers. These are individuals whose mental healthcare expenditures are, in any year in the pre-diagnosis period, above the 97.5th percentile of mental healthcare costs for that year, taking into account only consumers. Panel a shows employment for the people with HIV (solid blue line) and the matched controls (dashed gray line). Panel b shows the same for work hours, panel c for income and panel d for disability insurance utilization.

Figure R-2: Main outcomes - worst-case bounds outliers

Note: This figure shows mean outcomes after replacing the outcomes of individuals defined as mental healthcare expenditure outliers with worst-case bounds (0 for employment, work hours and income, 1 for DI). These are individuals whose mental healthcare expenditures are, in any year in the pre-diagnosis period, above the 97.5th percentile of mental healthcare costs for that year, taking into account only consumers. Panel a shows employment for the people with HIV (solid blue line) and the matched controls (dashed gray line). Panel b shows the same for work hours, panel c for income and panel d for disability insurance utilization.

2. Disability Insurance (DI): The argument presented in lines 417–429 lacks clarity. It is not evident that the analyses can rule out the possibility that the observed effects on employment, hours worked, and income are driven by DI.

One way to provide suggestive evidence would be to replicate the analyses on a subsample of individuals who never receive DI. If significant effects on employment and income persist within this subsample, it would strengthen the argument that the observed outcomes are not solely attributable to DI.

We thank the reviewer for this comment and acknowledge that the paragraph concerning disability insurance (DI) could be clarified. DI is an important institutional feature in the Netherlands, as individuals who stop working because of health reasons may qualify for income

through the DI system. It is for this reason that we study DI as a separate labor market outcome. It is important to acknowledge that individuals can qualify for partial DI in the Netherlands, implying that if they are sick, they may receive compensation based on a partial work capacity.

In our previous analysis, we found that that 36% of DI recipients are employed two years after diagnosis, which highlights that individuals with HIV may only receive partial DI, as shown in lines 432-434 of the manuscript. In a further analysis, we find that 23.4% of unemployed individuals receive DI two years after diagnosis, this share remains relatively constant in following years. These facts suggest that DI does not fully compensate for the decline in labor market participation after diagnosis.

In order to investigate to what extent the employment effects are driven by the availability of DI, we also ran the descriptive analysis on the subsample of individuals who are self-employed at time t-7. In the Netherlands, most self-employed individuals do not have access to the public disability insurance system but instead rely on self-paid private insurance. As many do not take out full insurance, the DI coverage rate is much lower among the self-employed¹. Therefore, the effect of HIV diagnosis on self-employment is much less likely to be explained by DI than the effect on employment. These results, included below in the Figure R3, show that there is a decrease in self-employment at the time of diagnosis. This implies that even individuals without access to public disability insurance experience a decline in labor market participation after diagnosis. We decided not to include this figure in the supplement but would of course be happy to do so if the editor/reviewer believes this is of added value.

In sum, our results suggest that while disability insurance access and utilization may be related to the other labor market outcomes, it does not fully explain the effects of an HIV diagnosis on employment, work hours and income.

We make the following edits to section 2.7:

Old (417 – 428): Another important mechanism that might explain the effects of an HIV diagnosis on labor market outcomes is the interaction between employment and disability insurance. In the Netherlands, individuals remain formally employed for the first two years of illness and continue being paid by their employers, after which their earnings capacity is assessed and they either enter a scheme for fully and permanently disabled individuals, or one for partly disabled individuals. Of the individuals with HIV in our sample who are receiving DI benefits two years after

¹ In 2020, the insurance rate was 17 percent for self-employed without personnel, and 30 percent for self-employed with personnel¹.

diagnosis, 78.8% have a degree of disability of at least 80%, 12.9% have a degree of disability between 45% and 80%, and 8.4% have a degree of disability of less than 45%. In addition, 36% of individuals who are receiving disability insurance or sickness benefits two years after diagnosis are also classified as employed for at least one month in the same year. Together, these figures suggest that, while disability insurance utilization may explain a part of the observed effect of an HIV diagnosis on income and employment, these effects are not entirely compensated by an increase in DI utilization.

New (423 – 437): *Finally, access to disability insurance (DI) represents an important institutional feature of the Dutch welfare system and labor market. In the Netherlands, individuals remain formally employed for the first two years of illness and continue being paid by their employers, after which their earnings capacity is assessed and they either enter a scheme for fully and permanently disabled individuals, or one for partly disabled individuals with an emphasis on reintegration into the labor market. While a majority (78.8%) of the individuals with HIV in our sample who are receiving DI benefits two years after diagnosis have a degree of disability of at least 80% (fully and permanently disabled) a relatively large share of 21.2% have lower degrees of disability that correspond to partial disability (12.9% have a degree of disability between 45% and 80%, and 8.4% have a degree of disability of less than 45%). Furthermore, 36% of individuals with HIV receiving DI or sickness benefits two years after diagnosis continue to be employed for at least one month, and a relatively small share of 23.4% of unemployed individuals are receiving benefits. Together, these numbers suggest that, while disability insurance utilization may explain a part of the observed effect of an HIV diagnosis on income and employment, these effects are not entirely compensated by an increase in DI utilization.*

We also make the following edit to section 3 Discussion:

Old (lines 505 – 508): *Second, the generalizability of our results might be limited to other high-income countries. The Netherlands provides a “best-case” scenario in which to measure the effects of HIV, as HIV care is widely available. Therefore, it is difficult to extrapolate these results to low- and middle-income settings.*

New (lines 513 – 518): *Second, the generalizability of our results might be limited to other high-income countries. The Netherlands provides a “best-case” scenario in which to measure the effects of HIV, as HIV care is widely available, including for individuals without access to health insurance. In addition, disability insurance is mandatory for wage workers. Therefore, it is difficult to extrapolate these results to low- and middle-income settings, or to settings with less generous welfare systems.*

Figure R-3: Self-employment and employment, conditional t-7

Note: This figure shows employment and self-employed over time relative to diagnosis. We condition on individuals being either employed or self-employed seven years prior to diagnosis. Panel a shows employment for people with HIV (solid blue line) and matched controls (dashed gray line), and panel b shows self-employment for these two groups.

References

1. ESB Redactie. Economisch Statistische Berichten Nummer 4805 . 48 Preprint at <https://esb.nu/wp-content/uploads/2023/07/Bladerpdf-4805-v2.pdf> (2022).